# Multimodal LLM-assisted Evolutionary Search for Programmatic Control Policies

**Qinglong Hu[1], Xialiang Tong[2], Mingxuan Yuan[2], Fei Liu[1], Zhichao Lu[1] & Qingfu Zhang[1]**
[1]Department of Computer Science, City University of Hong Kong, Hong Kong, China
[2]Huawei Noah's Ark Lab, China
`qinglhu2-c@my.cityu.edu.hk, qingfu.zhang@cityu.edu.hk`

## Abstract

Deep reinforcement learning has achieved impressive success in control tasks. However, its policies, represented as opaque neural networks, are often difficult for humans to understand, verify, and debug, which undermines trust and hinders real-world deployment. This work addresses this challenge by introducing a novel approach for programmatic control policy discovery, called Multimodal Large Language Model-assisted Evolutionary Search (MLES). MLES utilizes multimodal large language models as programmatic policy generators, combining them with evolutionary search to automate policy generation. It integrates visual feedback-driven behavior analysis within the policy generation process to identify failure patterns and guide targeted improvements, thereby enhancing policy discovery efficiency and producing adaptable, human-aligned policies. Experimental results demonstrate that MLES achieves performance comparable to Proximal Policy Optimization (PPO) across two standard control tasks while providing transparent control logic and traceable design processes. This approach also overcomes the limitations of predefined domain-specific languages, facilitates knowledge transfer and reuse, and is scalable across various tasks, showing promise as a new paradigm for developing transparent and verifiable control policies. Code is publicly available at `https://github.com/QingL2000/MLES`.

## 1 Introduction

High performance and transparency are two central challenges in the design of policies for control tasks (Milani et al., 2024). In recent years, deep reinforcement learning (DRL) has achieved remarkable performance across various domains. However, two fundamental challenges persist, stemming from the opaque, neural network-based nature of DRL policies (Vouros, 2022; Hickling et al., 2023; Puiutta & Veith, 2020). First, such policies function as black boxes, offering limited insight into their decision-making processes. This lack of transparency undermines trust, complicates verification, and hinders adoption, particularly in safety-critical applications such as autonomous driving and healthcare (Perez-Cerrolaza et al., 2024). Second, the policy learning process in DRL relies on gradient-based optimization over high-dimensional parameter spaces, making it difficult to analyze, intervene in, or reuse the learned knowledge. These limitations have driven the ongoing pursuit of policies that are not only high-performing but also transparent, verifiable, and human-readable.

The recent rise of large language models (LLMs) offers a promising opportunity to develop control policies that are both transparent and high-performing. LLMs possess strong capabilities in contextual understanding, reasoning, and generation, particularly excelling in coding-related tasks (Achiam et al., 2023). Building on this foundation, the *LLM-assisted Evolutionary Search* (LES) paradigm has emerged as a powerful framework for automated design. LES combines the generative and reasoning capabilities of LLMs with the iterative optimization strengths of Evolutionary Computation (EC)(Liu et al., 2023a). In this paradigm, LLMs serve as mutation or crossover operators, guided by prompt templates and evaluation feedback, to iteratively refine candidate designs (Romera-Paredes et al., 2024; Liu et al., 2024a; Ye et al., 2024). LES has shown success in the automated discovery of code, algorithms, and heuristics. In the reinforcement learning domain, recent studies such as *Eureka* et al. (Ma et al., 2024; Kwon et al., 2023; Sun et al., 2024; Masadome & Harada, 2025) have demonstrated that LES can be used to shape reward functions that improve agent performance.

Inspired by these advances, we pose the question of (1) whether LES can directly synthesize high-performance programmatic policies, rather than merely designing auxiliary components like reward functions. This direction holds the potential to yield agents that are both effective and transparent, thus bridging the gap between high-performance black-box models and human-understandable control logic. Building on this, we further investigate (2) how this process can be enhanced to generate more reliable policies, while also improving the efficiency of the policy discovery.

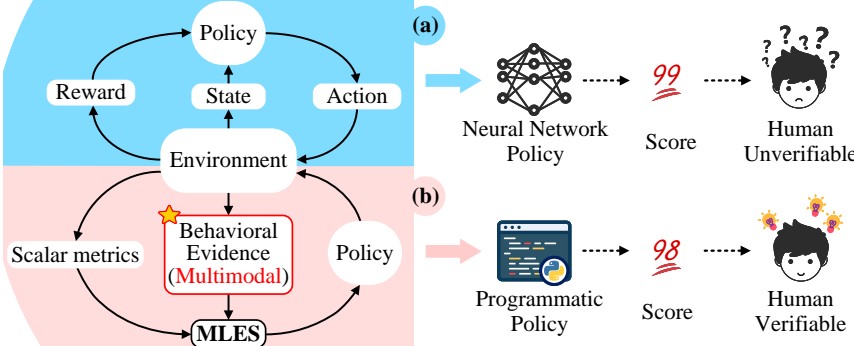

Figure 1: Paradigm comparison between DRL and MLES. (a) Standard DRL: agents learn opaque policies via reward-guided interaction with environments. (b) MLES: directly evolves programmatic policies by integrating behavior analysis into the EC-based policy discovery process.

To address these questions, we propose *Multimodal LLM-assisted Evolutionary Search* (MLES), a novel framework for the automated discovery of programmatic control policies. MLES advances the LES paradigm in two key ways. First, it targets the direct evolution of policies, where policies are expressed as executable programs with natural-language rationales. These are developed via natural language interactions with LLMs, enabling the synthesis of policies that are both semantically meaningful and easily understandable. Second, and most crucially, MLES introduces *visual feedback-driven behavior analysis* into the policy generation pipeline. By leveraging Multimodal LLMs (MLLMs) to scrutinize execution traces beyond mere scalar metrics, MLES enables targeted, rationale-driven policy refinement. This approach fundamentally transforms LLM-driven automated discovery from stochastic trial-and-error into a grounded, diagnostic refinement process, rendering the evolution transparent, traceable, and highly efficient. In summary, this paper makes the following contributions:

(1) We propose the general MLES framework, which integrates MLLMs with EC to directly synthesize programmatic control policies via environmental interaction. Unlike existing LES-based approaches that focus on reward function shaping, MLES enables end-to-end policy discovery, akin to DRL methods. A comparison between DRL and MLES is shown in Fig. 1.

(2) We present a prototypical instantiation of MLES and evaluate it on two standard RL benchmarks: Lunar Lander and Car Racing. Experimental results show that MLES produces effective policies with transparent control logic and traceable design processes, achieving performance comparable to a strong DRL baseline, Proximal Policy Optimization (PPO) (Schulman et al., 2017).

(3) We conduct an extensive analysis of MLES regarding its effectiveness and search efficiency, complemented by thorough ablation studies. We also investigate how different forms of behavioral evidence and prompt designs influence the policy evolution process.

## 2 METHODOLOGY

### 2.1 PROBLEM DEFINITION

Control policy discovery is the process of searching for high-quality policies within a predefined policy space to maximize expected performance in a given environment. Formally, given a policy space $\Pi$ and an evaluation function $F(\pi)$ that measures the quality of a policy $\pi \in \Pi$, the policy discovery problem can be formulated as the following optimization problem:

$$\pi^* = \arg\max_{\pi \in \Pi} F(\pi) \tag{1}$$

 Different policy discovery methods achieve this by varying the policy space $\Pi$ and the optimization mechanisms used. For instance, DRL methods employ optimizers to search for optimal neural network parameters within a continuous parameter space. Genetic Programming, combined with domain-specific languages (DSLs), uses evolutionary algorithms to explore and identify the optimal combinations of predefined grammars within a discrete grammar space.

In the case of MLES, policy discovery is framed as a search process driven by MLLMs within an open-ended semantic policy space. When given a control task, the boundaries and structure of the policy space are implicitly shaped by the MLLM's generative capabilities and world knowledge. This space is rich with task-relevant concepts and ideas, as well as valuable insights from other fields, allowing for flexible and expressive policy generation. By integrating MLLMs as policy generators within an evolutionary search process, MLES performs purposeful exploration and exploitation within this knowledge-rich space, facilitating the progressive discovery of high-performing policies.

## 2.2 THE GENERAL MULTIMODAL LLM-ASSISTED EVOLUTIONARY SEARCH FRAMEWORK

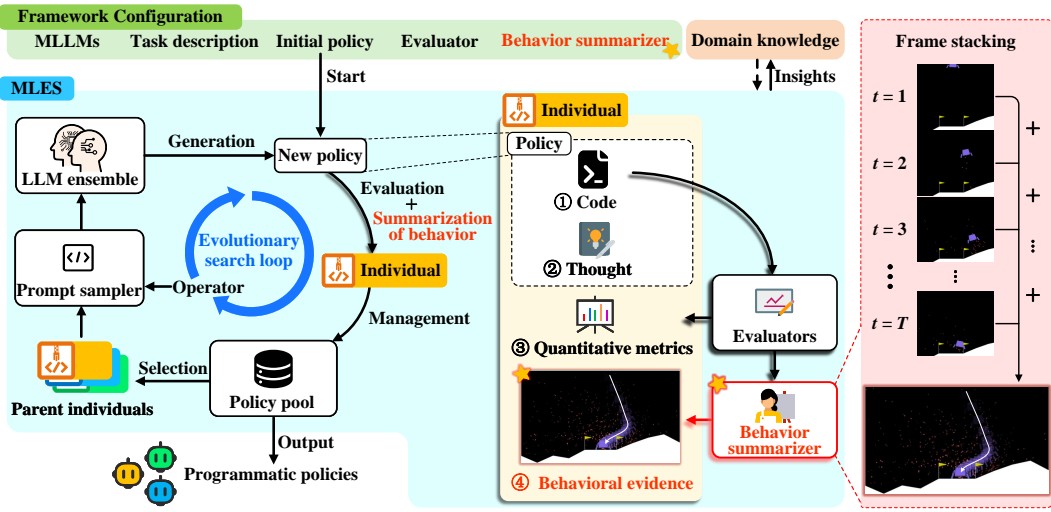

Figure 2: An overview of the MLES framework. The left side of the MLES illustrates the evolutionary search loop, while the right side details the structure and construction of an evolutionary individual. The red module on the far right exemplifies a behavior summarization pipeline (e.g., frame stacking) for generating behavioral evidence. At each search step, a subset of parent individuals is selected from the policy pool and used by the prompt sampler to create a multimodal few-shot prompt. MLLMs reason over this prompt to generate an offspring policy. The offspring is then evaluated and visualized, resulting in the creation of a new individual that is added to the policy pool and managed accordingly.

The MLES framework orchestrates a closed-loop evolutionary process to iteratively search for programmatic policies, as illustrated in Fig. 2. The framework requires the following configurable inputs: (1) a selection of MLLMs for policy generation, (2) a formal task description, (3) an initial set of programmatic policies (optional), (4) an evaluator for executing policies in the environment, and (5) a summarizer for generating behavioral evidence from policy executions. The framework consists of the following six core components:

**Evolutionary search loop.** The evolutionary search loop serves as the main control mechanism that orchestrates the entire evolutionary process. It manages iterations and maintains a dynamic balance between exploration and exploitation via configurable evolutionary operators, ensuring an effective search within the policy space.

**Policy pool.** It acts as a dynamic repository that holds the current policy population. It is responsible for selecting appropriate parent policies based on the evolutionary operator and integrating newly generated offspring into the pool, thereby facilitating the continuous evolution of the policies.

**Evaluators.** The evaluators are responsible for executing policies in the target environment. They output both quantitative performance metrics (e.g., episode rewards) and raw behavior traces. To improve efficiency and minimize delays, the policy evaluation procedure is often parallelized.

**Behavior summarizer.** It converts raw behavior traces collected by the evaluators into behavioral evidence. Unlike evaluators who merely quantify "how well" a policy performs, the summarizer provides rationale-rich feedback explaining "why" a policy succeeded or failed. This component is crucial for enabling informed, targeted policy modification, representing a key innovation of MLES.

**Prompt sampler.** It constructs multimodal few-shot prompts tailored to the current evolutionary operator by integrating selected parent policies and their corresponding behavioral evidence. These prompts provide essential context and guidance, directing the MLLMs to propose meaningful policies aligned with the intended transformation, thus steering the evolution toward promising directions.

**LLMs ensemble.** This component consists of a configurable set of MLLMs. These MLLMs reason over multimodal few-shot prompts, leveraging their advanced vision-language understanding, reasoning, and code generation capabilities to generate new candidate policies.

Through the iterative refinement process, MLES progressively discovers a diverse set of high-performing programmatic policies. Concurrently, this process explores a wide range of both successful and suboptimal policies, yielding valuable insights that enhance human understanding of the target task and provide a foundation for knowledge transfer and reuse in related control scenarios.

## 2.3 POLICY REPRESENTATION IN EVOLUTIONARY SEARCH PROCESS

During the discovery, each candidate policy is treated as an individual, structured as a tuple consisting of four elements that capture the policy's logic, intent, performance, and behavioral characteristics:

**Code.** Each policy is implemented as an executable Python program defining the agent's decision-making logic. This representation supports automated evaluation within the discovery and ensures direct compatibility with the environment. Moreover, expressing policies in code offers transparency and modularity, making them human-readable, easy to debug, and conducive to reuse or modification.

**Thought.** The thought is a concise natural language summary that captures the underlying rationale or design intent of a policy, highlighting key strategies that guide the agent's behavior. By providing a high-level semantic abstraction, thoughts help LLMs interpret the associated code more effectively and support deeper reasoning during policy generation. Prior work (Liu et al., 2024a) has shown that incorporating such descriptions into prompts can improve the efficiency and effectiveness of LES.

**Quantitative metrics.** Quantitative metrics are task-specific numerical evaluations obtained by executing the policy in the environment. These metrics serve as direct indicators of policy's performance (i.e., $F(\cdot)$ in 2.1), such as episode rewards, success rates, or other user-defined scores tailored to the tasks. They provide a consistent basis for ranking and selecting candidate policies, ensuring high-performing individuals are effectively identified and prioritized for further refinement.

**Behavioral Evidence. (BE)** While quantitative metrics measure performance, they provide only a partial understanding of a policy and often fail to offer actionable guidance for improvement. To address this limitation, MLES introduces BE, which captures the overall behavior of a policy or highlights key events that help identify areas for improvement. By analyzing BE, MLLMs can observe policy outcomes in detail, accurately identifying failure modes such as incorrect decisions or reward hacking (e.g., exploiting bugs for undeserved rewards). This enables more targeted modifications and corrections rather than merely blind trial-and-error, facilitating the discovery of human-aligned policies. This design mimics how human experts refine policies: not merely relying on numerical scores but also examining behavioral patterns (e.g., failure modes, unexpected interactions) to diagnose issues. The format of BE is flexible, including images, videos, and even text-based state sequences. Further examples illustrating the format and necessity of BE can be found in **Appendix D**.

Formally, MLES adopts a specialized division of labor: quantitative metrics act as the ground-truth objective for selection and convergence, while BE acts as the diagnostic signal for policy generation.

## 2.4 INSTANTIATION: THE MLES-EoH IMPLEMENTATION

### 2.4.1 THE EVOLUTIONARY BACKBONE

The pseudocode of the general MLES is shown in **Algorithm 1**. At each search step (Lines 4-7), the process iterates through parent selection, prompt construction, and offspring generation. By combining carefully designed operators with principled parent selection strategies, MLES achieves a balance between exploration and exploitation, enabling effective navigation of the policy space.

The MLES framework is agnostic to the underlying search algorithm. In this study, we instantiate it using the established Evolution of Heuristics (EoH) (Liu et al., 2024a) as the evolutionary backbone. EoH uses natural-language "Thought" to elicit LLM reasoning, thereby guiding coherent search over the heuristic space. In our instantiation (referred to as MLES-EoH, hereafter simply MLES), we integrate *visual feedback-driven behavior analysis* into the policy generation pipeline to discover high-performing programmatic policies. We deliberately minimize structural changes to the search operators and maintain a consistent evolutionary loop. This design provides a controlled experimental platform that rigorously isolates the impact of visual feedback-driven behavior analysis, distinguishing our contribution from performance gains that might otherwise stem from complex genetic operators or search strategies.

**Parent selection.** (Line 4) Parent policies are selected from the policy pool using *exponential rank selection*, a standard method in EC (Blickle & Thiele, 1996). In this scheme, policies are ranked by performance, and the selection probability $p_i$ for the policy at rank $r_i$ is proportional to an exponential decay of its rank, typically expressed as $p_i \propto e^{-c \cdot r_i}$. Unlike traditional evolutionary algorithms, where mutation is cheap, MLES relies on MLLM inference to generate offspring. Therefore, maintaining high selection pressure is critical for sample efficiency. This scheme prioritizes high-performing individuals for exploitation while preserving a non-zero probability for lower-ranked policies to aid exploration, ensuring diversity is not prematurely lost.

---

**Algorithm 1** Pseudocode of MLES

**Require:** Task description, training instances, interfaces of Evaluator and Summarizer
1: Initialize policy pool
2: **while** termination condition not met **do**
3:     **for** each evolutionary operator **do**
4:         Select parent policies from policy pool
5:         Obtain *BE*s of parent policies
6:         Construct multimodal few-shot prompt
7:         Generate offspring policy via MLLM
8:         Evaluate offspring and create its *BE*
9:         Manage and update policy pool
10:     **end for**
11: **end while**
12: **return** A pool of control policies

---

**Policy pool management.** (Line 9) Upon generation and evaluation, offspring are filtered for redundancy. To maintain population quality, the top $N$ individuals, ranked by quantitative metrics, are retained for the next generation, facilitating progressive refinement.

### 2.4.2 POLICY GENERATION ENHANCED BY MULTIMODAL BEHAVIORAL ANALYSIS

**Evolutionary operators.** MLES employs a set of prompting instructions (i.e., operators) to direct the MLLM's generation process. We adapt the standard exploration operators from EoH and introduce novel multimodal modification operators:

- Exploration operators (E1 and E2): These operators aim to enhance the population's behavioral diversity by generating novel policies based on the analysis of parent policies' control logic. **E1** prompts the LLM to examine both the code and the thought of selected parent policies, then to synthesize a new policy that implements fundamentally different control strategies, encouraging exploration of uncharted regions of the policy space. **E2** prompts the LLM to identify shared patterns across multiple parent policies and construct a new policy that generalizes from these patterns, akin to the crossover operation in evolutionary computation.

- Multimodal modification operators (M1_M and M2_M): These operators instruct MLLMs to leverage BE to refine existing policies through detailed, informed modification. **M1_M** directs the MLLMs to identify behavioral shortcomings by jointly analyzing the policy code and its BE, and subsequently revise the control logic accordingly. **M2_M** prompts the MLLMs to identify critical parameters in the policy and adjust them based on observed evidence, thereby enabling

targeted optimization. The integration of BE analysis provides grounded insights for the MLLM's reasoning, resulting in more coherent and purposeful offspring generation.

**Multimodal few-shot prompt construction.** (Line 6) Given an operator and selected parents, the prompt sampler dynamically assembles a four-part context: (1) static task description, (2) parent information (code & thought), (3) relevant BEs, and (4) operator-specific instructions. This composite prompt enables the MLLMs to function not just as coders but as reasoning agents that observe, diagnose, and improve.

**Policy generation & individual construction.** (Line 7 & 8) Guided by the constructed prompts, the MLLM generates a candidate policy. The policy is executed by evaluators to assess the quantitative metrics. After that, the raw data from its executions is processed by the behavior summarizer, which converts it into BE and ultimately leads to the construction of the individual. This BE is stored and directly reused when the individual is selected as a parent in future generations (Line 5).

# 3 EXPERIMENTS

## 3.1 EXPERIMENTAL SETTING

**Benchmarks.** We evaluate the proposed MLES framework on two representative control tasks from the OpenAI Gym suite (Brockman et al., 2016): **Lunar Lander** and **Car Racing**. These tasks collectively cover both discrete and continuous control settings, providing a comprehensive testbed for our method.

- **Lunar Lander** is a discrete-action control task where the agent must land a spacecraft on a designated pad. The agent has four action choices, and the environment has an 8-dimensional state space. It is commonly used for benchmarking RL methods in discrete action domains.
- **Car Racing** is a more complex continuous-control task. The agent drives a car along procedurally generated tracks based on pixel observations. This control task poses a greater challenge due to its high-dimensional input ($96 \times 96 \times 3$ image) and continuous action space, making it an ideal testbed for evaluating image-conditioned programmatic policies.

**Baselines.** We compare MLES against both DRL methods and an existing LES approach for direct control policy discovery:

- **Deep Q-Network** (DQN) (Mnih et al., 2013) is a value-based DRL algorithm designed for discrete action spaces. It serves as a representative DRL baseline for Lunar Lander.
- **Proximal Policy Optimization** (PPO) (Schulman et al., 2017) is a widely used on-policy policy gradient method that can handle both discrete and continuous control tasks.
- **EoH** (Liu et al., 2024a) is a recent LES framework for automated code discovery. We use it as a baseline to assess the added value of behavioral evidence analysis in MLES, as it shares the same evolutionary framework but lacks the visual feedback-driven behavior analysis used in MLES.

Beyond the baselines listed above, we also considered other paradigms for generating human-readable policies, such as genetic programming (GP) and policy distillation. However, GP-based approaches are highly dependent on manually-designed DSLs (Trivedi et al., 2021; Qiu & Zhu, 2022). Adapting them to our benchmarks would require non-trivial effort, making a fair comparison difficult. Regarding policy distillation, as a post-hoc approach that fits an interpretable surrogate to a pre-trained oracle, it fundamentally differs from our goal of directly discovering policies through environmental interaction (similar to DRL). Furthermore, it introduces computational disparities that are difficult to reconcile, as it necessitates the pre-training of high-performance black-box policies followed by a rigorous distillation protocol. Therefore, evaluating MLES against high-performing DRL methods is the most direct and meaningful measure of its capabilities.

**Implementation details.** Both MLES and EoH employ GPT-4o-mini for policy generation. The LLM query budget is set to 2000 requests, corresponding to 10,000 and 8,000 environment resets for Lunar Lander and Car Racing, respectively. To ensure a fair comparison, we train all baselines under an identical budget of environment resets. The policy pool size $N$ is set to 16, with the initial population of policies for MLES shared with EoH to avoid any biases arising from different starting points. For the Lunar Lander task, we select five representative instances for training, while for the

Car Racing task, four representative tracks are chosen. Both tasks use ten test environments, each generated from a random seed in the range zero to nine. Each experiment is repeated five times to ensure the robustness and reliability of the results.

## 3.2 EXPERIMENTAL RESULTS

Table 1: Performance on benchmarks. Best results are in **bold**, second-best are underlined.

| Method | Lunar lander | | | | Car Racing | | | |
|--------|---------|----------------|----------|----------------|---------|----------------|----------|----------------|
| | | Training | | Testing | | Training | | Testing |
| | Average | SEM | Best run | Average | Average | SEM | Best run | Average |
| DQN | 1.017 | ±0.022 | 1.066 | 0.508 | 79.777 | ±2.635 | 91.182 | 71.724 |
| PPO | 1.032 | ±0.026 | 1.076 | **0.846** | **99.212** | ±0.390 | **100.000** | 94.546 |
| Initial policy | 0.629 | — | — | 0.653 | 17.772 | — | — | 17.619 |
| Eoh | 1.053 | ±0.021 | 1.085 | 0.776 | 89.808 | ±1.553 | 96.675 | 79.286 |
| MLES | **1.090** | **±0.005** | **1.098** | 0.819 | 98.070 | ±0.719 | **100.000** | **96.358** |

**Comparative evaluations of MLES.** Table 1 presents the performance of each method on Lunar Lander and Car Racing benchmarks. We report the average and best performance across five independent runs for both training and testing, along with the standard error of the mean (SEM). On the Lunar Lander task, a score above 1.00 signifies a successful landing on all instances, with higher scores indicating greater fuel savings. For the Car Racing task, a score of 100 represents a flawless completion of all tracks. For both tasks, a higher score indicates better performance. The specific metrics used for these evaluations are detailed in **Appendix C**.

As Table 1 shows, the MLES achieves the highest training performance on Lunar Lander and ranks second-best on Car Racing, performing comparably to the strong PPO baseline. Compared to the Initial Policy, which serves as our starting point, MLES demonstrates substantial improvements across both tasks, highlighting its practical effectiveness. These results provide strong evidence that MLES can directly synthesize high-performing policies through environment interaction, achieving performance on par with strong DRL baselines.

When compared to EoH, MLES consistently and significantly outperforms it on both training and testing instances. Furthermore, MLES exhibits superior convergence and stability, as indicated by its smaller SEM. The performance gap is especially pronounced in the more challenging Car Racing task. This outcome demonstrates that integrating behavior analysis not only substantially enhances the effectiveness of policy evolution but also improves the robustness of the generated policies.

However, we observe a consistent drop in testing performance relative to training across all methods. This indicates that generalization remains a challenge. In **Appendix B.6**, we explore how MLES can leverage its population-based nature to improve generalization through policy ensembling.

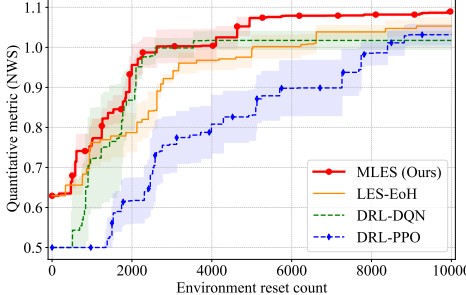
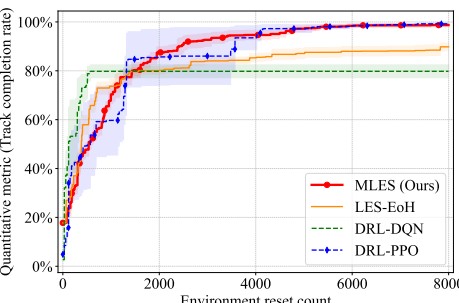

Figure 3: Convergence on Lunar Lander task   Figure 4: Convergence on Car Racing task

**Analysis of policy discovery efficiency.** Figs. 3 and 4 illustrate the convergence of the best policy performance with respect to the count of environment resets for four methods. Colored curves represent the mean performance over five runs, and shaded areas indicate the SEM, reflecting the stability during the policy discovery process.

As demonstrated, the MLES consistently shows superior search efficiency compared to the baselines. On the simpler Lunar Lander task, MLES discovers high-performing policies within approximately 5,000 environment resets, which is substantially faster than both PPO and DQN. On the more challenging Car Racing task, MLES achieves a convergence speed comparable to PPO while exhibiting significantly lower variance across runs, indicating more stable learning dynamics. Compared with EoH, MLES achieves faster convergence and higher final performance across tasks. The narrower confidence intervals in later stages further emphasize MLES's robustness and consistency. These results highlight the benefits of incorporating visual feedback-driven behavior analysis into the evolutionary process. By leveraging richer signals beyond scalar performance metrics, MLES more effectively guides policy search, leading to faster convergence and enhanced stability.

Another noteworthy observation is that both MLES and EoH achieve higher initial performance than DRL baselines. This advantage stems from the ability of the LES paradigm to readily leverage prior knowledge as initial policies, providing a more informed starting point for the search. In contrast, defining meaningful policies in the form of neural networks is challenging, and DRL methods typically rely on randomly initialized weights. This highlights MLES's inherent capacity to reuse and transfer knowledge. This capability is particularly valuable in domains where expert heuristics are available, enabling more sample-efficient exploration from the outset.

**Qualitative analysis of the transparency.** Fig. 5 presents an evolutionary process for Car Racing policies, showing how individual scores change over generations. This visualization clarifies why each policy was generated and how it was refined. Such transparency firmly demonstrates that the MLES policy discovery process is interpretable and traceable. By analyzing the BE, the MLLMs identify failure patterns in parent policies and perform targeted improvements accordingly. The final policy's code and accompanying thought are provided in **Appendix F**. To validate the human-readability of the generated policies, we asked 20 graduate students with computer science backgrounds to review them. The participants reported that they could understand the code and noted that the detailed comments embedded within the code were particularly helpful. In summary, MLES offers transparency on two levels: (1) the discovered programmatic policies are fully human-readable; (2) the policy discovery process is entirely transparent and highly informative for further research.

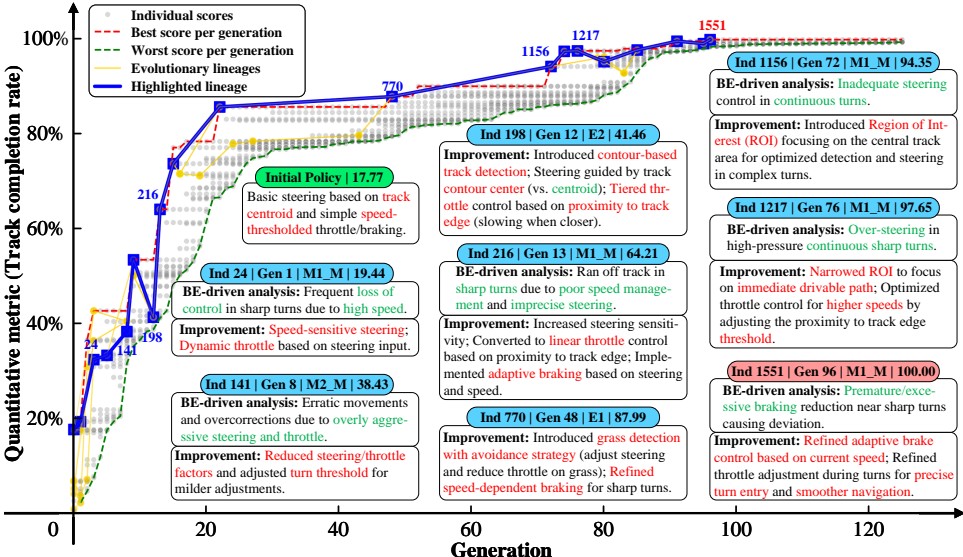

Figure 5: Evolutionary process of Car Racing policies. The plot depicts population score distributions over generations, with yellow lines tracing all ancestors of the best-performing policy. The blue lineage is examined in detail to reveal the stepwise improvements guided by BE-driven insights.

**Verification of cold-start capability.** Table 2 presents the performance of MLES when starting entirely from scratch (i.e., w/o initial policy) under a budget of 2,000 LLM queries. As observed, although the cold-start setting inherently requires more exploration to bootstrap the search, MLES remains fully capable of autonomously discovering high-performing policies. Notably, when powered by the more advanced GPT-5-mini, MLES achieves strong performance even without seeds,

successfully solving tasks devoid of human prior knowledge within the constrained query budget. These results validate MLES's robustness in cold-start scenarios and its adaptability to novel tasks (see **Appendix B.5** for additional experiments on more control tasks). Moreover, the significant improvement observed in seeded runs using the same MLLM confirms that ingesting prior knowledge accelerates the policy discovery process. This highlights MLES's capability and flexibility in leveraging existing knowledge to jump-start optimization in domains with established baselines.

Table 2: Analysis of MLES's dependence on initial priors and MLLM capabilities.

| Experimental Setting | | Lunar Lander | | Car Racing | |
| --- | --- | --- | --- | --- | --- |
| Initial Policy | MLLM | Score @ 2k | Samples to Match[†] | Score @ 2k | Samples to Match[†] |
| w/ | GPT-4o-mini | **1.090** | *(Baseline)* | 98.07 | *(Baseline)* |
| | GPT-5-mini | 1.065 | >2000 | **100.00** | 346 |
| w/o | GPT-4o-mini | 0.667 | *(Baseline)* | 91.96 | *(Baseline)* |
| | GPT-5-mini | 1.013 | **258** | **100.00** | **209** |

*Note:* **Score @ 2k** denotes the final performance achieved after 2,000 MLLM queries. **Samples to Match**[†] indicates the number of queries required for GPT-5-mini to match (or exceed) the final performance of the GPT-4o-mini baseline within the same initialization setting; lower values indicate higher search efficiency. ">2k" denotes failure to match the baseline score within the 2,000-query budget.

**Impact of underlying MLLM capabilities.** Table 2 compares the performance of MLES using different MLLMs under identical settings. As illustrated, employing stronger MLLMs significantly boosts both performance and search efficiency. For instance, on the Car Racing task, GPT-5-mini achieves the maximum score (100.0) using only $\sim$17% of the sample budget required by GPT-4o-mini. This suggests that MLES will naturally gain efficiency and power as foundation models continue to advance, demonstrating the framework's longevity and scalability.

## 3.3 ABLATION STUDY ON EVOLUTIONARY OPERATORS

We conduct an ablation study to assess the individual contributions and the collective synergy of the evolutionary operators within the MLES (Table 3, with more detailed results and analysis provided in **Appendix B.4**). The results presented are based on three independent runs, and the metric $\Delta$Perf(%) quantifies the performance degradation relative to the full MLES framework.

The most pronounced performance degradation is observed in the variants that remove the multimodal modification operators (w/o M1_M and w/o M2_M). This highlights the critical role of leveraging concrete execution feedback to rectify behavioral shortcomings and fine-tune parameters, which significantly enhances the efficiency of the policy discovery process. In contrast, the exploration operators

Table 3: Ablation study results on two control tasks

| Method | Lunar Lander | | Car Racing | |
| --- | --- | --- | --- | --- |
| | Average | $\Delta$Perf (%) | Average | $\Delta$Perf (%) |
| MLES | 1.090 | 0.00% | 98.70 | 0.00% |
| w/o E1 | 1.093 | +0.17% | 87.53 | -11.32% |
| w/o E2 | 1.082 | -0.68% | 89.04 | -9.79% |
| w/o M1_M | 0.997 | -8.54% | 86.71 | -12.15% |
| w/o M2_M | 1.020 | -6.41% | 84.77 | -14.12% |

(w/o E1 and w/o E2) exhibit different impacts depending on task complexity. Their absence leads to a notable decrease in performance on the more challenging Car Racing task, suggesting that the exploration capabilities of E1 and E2 become increasingly vital in complex policy spaces. However, the effect on the simpler Lunar Lander is minimal, indicating that in less complex environments, their aggressive exploration might introduce counterproductive variance.

In summary, the full MLES outperforms all ablated variants, confirming the synergistic effect of the operators. The exploration operators serve as the "engine of creativity," broadening the search by introducing diverse policies into the population. Subsequently, the multimodal modification operators act as the "engine of refinement," grounding the search by improving these policies based on behavioral evidence. This powerful combination allows MLES to effectively balance a broad, creative policy search with a deep, evidence-driven refinement of promising solutions.

## 4    RELATED WORKS

**Learning human-readable policies.** Existing methods for creating transparent policies can be broadly categorized as post-hoc and direct approaches (Vouros, 2022; Glanois et al., 2024). Post-hoc methods aim to explain a pre-trained black-box policy by distilling its behavior into a more transparent model, such as a decision tree (Bastani et al., 2018; Gokhale et al., 2024; Kohler et al., 2024), a mathematical formula (Hein et al., 2018; Landajuela et al., 2021; Hazra & De Raedt, 2023), or a program in a DSL (Verma et al., 2018; Verma, 2019). Although they can achieve high performance by mimicking black-box policies, they provide limited insight into proactively constructing or improving policies, and tend to lack generalization beyond the oracle's domain (Gu et al., 2025). In contrast, direct approaches discover a policy from scratch through interaction with the environment. This category includes methods that generate policies as logic rules (Jiang & Luo, 2019; Delfosse et al., 2023; Glanois et al., 2022) or programs within a pre-defined DSL (Qiu & Zhu, 2022; Liu et al., 2023b; Gu et al., 2024; Liu et al., 2024b). However, they are often constrained by static grammars or limited search spaces, restricting their expressiveness and scalability. Our MLES falls into the latter category but significantly expands its scope by representing policies as executable programs with natural language explanations and using pre-trained LLMs to generate more expressive and diverse control logic, without relying on expert data or static, handcrafted grammars.

**LLM-assisted evolutionary search.** Recent advances in EC have integrated LLMs into the evolutionary loop, giving rise to a new paradigm known as LES (Liu et al., 2023a). This paradigm has shown success in areas such as code generation (Hemberg et al., 2024), scientific discovery (Romera-Paredes et al., 2024; Shojaee et al., 2024), and algorithm design (Liu et al., 2024a; Ye et al., 2024). In the RL domain, LES has been primarily applied to the indirect optimization of agents by evolving reward functions (Ma et al., 2024; Hazra et al., 2024), thereby improving the performance of downstream neural policies. However, the resulting policies remain neural and opaque. Our MLES extends LES toward the direct synthesis of programmatic policies through interactions with the environment. Additionally, we incorporate behavior analysis into the evolutionary loop, enabling the search to be guided by richer information beyond scalar rewards. This facilitates a more informed and transparent form of policy discovery, where both the resulting policy and its design rationale are accessible for inspection. To the best of our knowledge, this is the first effort to unify LLMs, evolutionary search, and multimodal evaluation for the direct synthesis of policies in control tasks.

**Multimodal large language models.** MLLMs extend the capabilities of LLMs by integrating visual and textual modalities, enabling grounded understanding and vision-language reasoning (Caffagni et al., 2024). Recent models such as GPT-4V (Hurst et al., 2024) and Qwen2.5-VL (Bai et al., 2025) demonstrate strong performance on tasks requiring joint visual-language comprehension for planning and high-level control. These advances have led to increasing applications of MLLMs in planning and optimization (Zheng et al., 2024; Szot et al., 2025; Elhenawy et al., 2024a;b; Zhao & Cheong, 2025). More recently, MLLMs have also been used to adapt reward functions based on visual cues (Narin, 2024; Wang et al., 2025; Cuzin-Rambaud et al., 2025), thereby improving agent learning in complex RL settings. Our MLES uses the vision-language capabilities of MLLMs to guide the policy search process with rich information, including policy execution traces, in-depth analyses, and promising corrective suggestions. This approach facilitates more transparent, traceable policy discovery, ultimately supporting the synthesis of reliable, human-aligned policies.

## 5    CONCLUSION

This paper introduces Multimodal Large Language Model-assisted Evolutionary Search (MLES), which combines multimodal large language models with evolutionary computation to efficiently design programmatic policy for control tasks. By incorporating visual feedback-driven behavior analysis, MLES mimics a human-like process for policy development, thereby enhancing search efficiency. Our experiments on two standard control benchmarks demonstrate that MLES generates effective policies with performance comparable to the strong Proximal Policy Optimization (PPO) baseline. Notably, MLES yields transparent control logic with traceable policy design processes. MLES offers a flexible, automated approach that reduces human effort and facilitates knowledge reuse, presenting a promising new paradigm for developing transparent, verifiable control policies.

## REPRODUCIBILITY STATEMENT

The technical details of the MLES framework are provided in Section 2. To ensure full reproducibility of our experimental results, we have included all necessary information in the supplementary materials. This includes a detailed breakdown of all hyperparameters and experimental setups in Appendix B.1, and a comprehensive list of the prompts used in Appendix E. Furthermore, the source code is publicly available at `https://github.com/QingL2000/MLES` to support the replication of all reported experiments and findings.

## ACKNOWLEDGMENTS

This work was supported by the General Research Fund (GRF) of the Research Grants Council (RGC) of Hong Kong (Project No. CityU 11217325).

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

# A DISCUSSION AND LIMITATION

## A.1 OUTLOOK OF MLES FOR CONTROL POLICY DISCOVERY

This paper provides a preliminary yet compelling demonstration of the Multimodal Large Language Model-assisted Evolutionary Search (MLES) framework for automatically discovering programmatic control policies. As evidenced by the results in the Car Racing task (Appendix F), MLES successfully constructed a data processing pipeline for image input, guiding control decisions for steering, throttle, and braking with flawless performance across various tracks. This foundational success points toward several exciting future directions and highlights the framework's significant potential.

**Modularity enables hybrid architectures for complex scenarios.** As observed in Appendix F, the programmatic policies discovered by MLES are inherently modular, typically separating into feature processing and decision-making components. This separation provides a powerful pathway for tackling more complex environments. Our framework's ability to autonomously leverage sophisticated libraries such as OpenCV underscores its potential to integrate even more advanced perception modules.

For instance, a pre-trained convolutional neural network (CNN) could be employed for high-dimensional visual processing. The output from this module—whether processed features or inter-mediate computations—can then be fed into a transparent control logic for decision-making. This creates a compelling opportunity for hybrid AI systems that merge the feature-learning power of deep learning with the transparency and verifiability of code-based control logic. In stark contrast to end-to-end DRL, where perception and control are inextricably coupled within an opaque network, our approach ensures that even when perception is a black box, the critical decision logic remains fully transparent and verifiable. Such a capability is not just an advantage but a necessity for real-world applications like autonomous driving and embodied intelligence, where handling complex sensory inputs must be paired with the highest levels of safety and verifiability.

**Human-in-the-loop collaboration.** The modular, code-based nature of MLES policies also uniquely facilitates human-expert collaboration. In DRL, the tight coupling of the entire network makes it challenging to modify specific control preferences. MLES, however, allows experts to inspect and intervene at any stage of the policy discovery process. If a deficiency is identified, an expert can manually edit the relevant code block and re-insert the improved policy into the evolutionary pool for further automated refinement. This enables continuous expert-assisted refinement. Furthermore, this collaboration can be streamlined via natural language: an expert could simply identify a failure mode (e.g., "the agent brakes too late on sharp turns") and provide this feedback in a prompt, allowing the MLES framework to automatically generate a targeted modification. This creates a seamless and efficient workflow for expert-guided policy optimization.

**Facilitating knowledge transfer and reuse.** One of the most significant advantages of MLES is that the discovered policies are encoded as executable code, which greatly simplifies knowledge transfer and reuse. In DRL, transferring knowledge often involves complex mechanisms like transfer learning or fine-tuning, which can be resource-intensive and difficult to implement. MLES bypasses these challenges by directly encoding prior knowledge into code. This approach not only makes knowledge transfer easier but also accelerates policy deployment. Additionally, LLMs can assist in this translation process, further reducing the time and effort required. As a result, MLES is particularly beneficial for tackling problems that involve well-established expert knowledge.

Moreover, this ease of transfer extends to addressing generalization challenges. In many tasks, different instances of the same problem may require subtle adjustments to the policy for optimal performance. Despite these variations, the policy's core logic generally remains consistent. As such, instead of starting from scratch for each new instance, MLES can adapt and fine-tune existing policies with minimal adjustments, ensuring continued high performance. A promising future direction for MLES is the collaborative evolution of policies for instances across multiple distributions, facilitating knowledge sharing of high-performing policies across different scenarios during the evolutionary process. This approach will be demonstrated in future work.

By integrating visual feedback-driven behavior analysis, MLES mimics how human experts design policies—actively evaluating and adjusting them in response to feedback. The key advantage is

that MLES operates fully automatically and can scale indefinitely, given sufficient computational resources. This scalability significantly accelerates the discovery of viable policies. In summary, MLES has substantial potential as a promising paradigm for transparent control policy discovery. We believe that MLES offers a new path forward for the field and invite the broader research community to explore and contribute to this exciting area.

## A.2 COMPUTATIONAL CONSIDERATIONS AND COST-BENEFIT ANALYSIS

MLES introduces a computational paradigm distinct from traditional DRL. Instead of relying on intensive and prolonged local GPU computation for model training, the primary resource consumption in MLES stems from the inference process of Multimodal Large Language Models (MLLMs).

This inference can be realized through two primary modalities: API-based services or local deployment. In this work, we utilize the API-based approach. While this incurs a direct financial cost, the expenditure is surprisingly modest, especially given the unique value it generates. For instance, our experiments show that solving the Car Racing benchmark, which required approximately 2000 requests to a cost-effective MLLM (such as GPT-4o mini), incurred a total cost of around 0.42 USD. Crucially, this nominal cost achieves a goal fundamentally unattainable by DRL, regardless of its computational budget: the generation of transparent, human-readable programmatic policies. Therefore, from a cost-benefit perspective, MLES offers an exceptionally favorable trade-off, exchanging a minor financial outlay for a critical capability that traditional methods cannot provide.

Furthermore, this API-centric paradigm offers a significant, often-overlooked advantage: a minimal requirement for local hardware resources. The computationally demanding MLLM inference is offloaded to a third-party service, removing the need for powerful local GPUs and thus lowering the barrier to entry for researchers and practitioners. For institutions with the capacity for local MLLM deployment, this direct financial cost is effectively eliminated, reducing the expenditure to local inference costs alone.

Given the continuous downward trend in LLM API pricing and the proliferation of powerful open-source models, we believe the MLES paradigm is not only viable but also a highly practical and sustainable approach for developing the next generation of trustworthy AI control systems.

## A.3 LIMITATION

**Task-specific design of behavioral evidence.** The MLES framework leverages Behavioral Evidence (BE) to analyze policy behavior and guide targeted improvements, thereby accelerating policy discovery. A current limitation, however, is that the design of efficient BE remains a task-specific, manual process. For MLES to achieve optimal performance, the BE must be sufficiently informative to capture the nuances of policy failures. In this paper, we design BEs from a human perspective for the Lunar Lander and Car Racing tasks and successfully apply them within MLES, achieving high performance. However, for a new task, the design of an appropriate BE-generation pipeline is required, introducing design overhead.

We foresee three promising avenues to reduce or eliminate this manual effort:

- **First**, by reusing and fine-tuning the BE-generation pipeline proposed in this paper. As detailed in Appendix D.2, methods such as frame stacking are well-suited for tasks with relatively static backgrounds, while trajectory mapping is effective for tasks focused on object motion, such as vehicle or drone control. These pipelines can serve as robust starting points for new environments.

- **Second**, by leveraging video-format BE, such as raw video clips of agent rollouts. This approach would provide rich, detailed information with minimal design effort but introduces a trade-off with computational cost, primarily due to the high token count associated with processing image sequences in MLLMs. The feasibility of this approach will likely improve as the cost of MLLM inference decreases over time.

- **Third**, a more ambitious direction is to empower the MLES framework to automatically design its own BE. Analogous to the way MLES designs programmatic policies, an outer design loop could be added to the framework to optimize the BE-generation pipeline itself. This can be framed as a meta-learning problem, where the outer loop discovers an optimal BE pipeline that provides

maximally informative evidence for the inner policy evolution loop, thereby minimizing the need for human intervention.

**Dependency on multimodal large language models.** The performance ceiling of MLES is inherently coupled with the capabilities of its underlying MLLMs. The framework's ability to understand complex failure modes depends on the MLLM's visual reasoning, while the quality of generated policies depends on its code-generation and logical-deduction abilities.

While this dependency is a limitation, it also means that MLES is designed to continuously improve as foundational models advance. As MLLMs become more capable, the performance, efficiency, and scalability of MLES will naturally be enhanced. This symbiotic relationship positions MLES not as a static method, but as an evolving paradigm poised to leverage future breakthroughs in generative AI. Future work could also explore systematically evaluating the impact of different open-source and proprietary MLLMs within the MLES framework to better understand this relationship.

# B  ADDITIONAL EXPERIMENT RESULTS

## B.1  HYPERPARAMETERS FOR EXPERIMENTAL SETUP

This subsection presents the hyperparameters used in our experiments involving MLES, EoH, DQN, and PPO. The experiments were implemented in Python and executed on a single CPU (Intel i9-13980HX) with 32GB of RAM. Table 4 details the hyperparameters for MLES and EoH, where identical settings were applied to ensure a fair comparison, based on the configurations from the EoH paper (Liu et al., 2024a). Tables 5 and 6 present the hyperparameter settings for DQN and PPO across two tasks, respectively. Tables 7 and 8 show the network architectures used for DQN and PPO on two benchmarks. To ensure a fair comparison, we maintained identical network structures (with approximately equal parameter counts) for DQN and PPO in terms of feature processing and decision-making, excluding the output layers.

Notably, while the PPO agent utilizes a densely parameterized neural network (as shown in Table 8), the programmatic policies discovered by MLES achieve comparable performance with a significantly lower computational footprint. This stark difference in complexity makes MLES policies exceptionally well-suited for deployment in real-world applications, particularly in resource-constrained environments such as embedded systems or micro-controllers.

Table 4: Hyperparameter settings for MLES and EoH.

| Hyperparameter | Parameter Description | Value |
|---|---|---|
| $m$ | Number of parents selected in each evolution step | 2 |
| $N$ | Population size | 16 |
| LLM | Version of LLM used in the evolutionary operators | GPT-4o-mini |
| LLM temperature | Hyperparameter controlling the randomness of LLM text generation | 1 |

Table 5: DQN Hyperparameters for Lunar Lander and Car Racing Tasks

| Hyperparameter | Lunar Lander | Car Racing |
|---|---|---|
| Action Space Dimension | 4 | 9 (Discrete actions) |
| Replay Buffer Capacity | 10,000 | 30,000 |
| Batch Size | 128 | 128 |
| Discount Factor | 0.99 | 0.98 |
| Initial $\epsilon$ | 1.0 | 1.0 |
| Minimum $\epsilon$ | 0.01 | 0.1 |
| $\epsilon$ Decay Rate | 0.9995 | 0.995 |
| Target Network Update Rate ($\tau$) | 0.01 | 0.01 |
| Optimizer Type | Adam | Adam |
| Learning Rate | 0.001 | 0.0005 |

Table 6: PPO Hyperparameters for Lunar Lander and Car Racing Tasks

| Parameter | Lunar Lander | Car Racing |
|---|---|---|
| Discount Factor | 0.99 | 0.99 |
| Generalized Advantage Estimation Parameter | 0.95 | 0.95 |
| PPO Clip | 0.2 | 0.2 |
| Entropy Coefficient | 0.01 | 0.01 |
| Critic Coefficient | 0.5 | 0.5 |
| Max Gradient Norm | 0.5 | 0.5 |
| Batch Size | 256 | 256 |
| Mini Batch Size | 64 | 64 |
| Learning Rate | 3e-5 | 3e-5 |

Table 7: Network Architectures for DQN and PPO on Lunar Lander Task

| Component | Layer Type | DQN Configuration | PPO Configuration |
|---|---|---|---|
| Input Layer | State Dimension | 8 (environment observation space) | |
| Hidden Layer | Layer 1 | Linear(512)→ReLU | |
| | Layer 2 | Linear(512)→ReLU | |
| Output Layer | Policy Head | Linear(4) | |
| | Value Head | N/A | Linear(1) |
| Activation | Hidden Layers | ReLU | |
| | Output Layer | Linear | |
| Parameters | Total Parameters | 269,316 | 269,829 |

Table 8: Network Architectures for DQN and PPO on Car Racing Task

| Component | Specification | DQN | PPO |
|---|---|---|---|
| Input Layer | Dimensions | 96×96×3 (Current frame) + 96×96×3 (Last frame) + 1 (Speed) + 3 (Last action) | |
| Convolutional Layers | Layer 1 | Conv2d(3→32, kernel=8×8, stride=4) → ReLU | |
| | Layer 2 | Conv2d(32→64, kernel=4×4, stride=2) → ReLU | |
| | Layer 3 | Conv2d(64→64, kernel=3×3, stride=1) → ReLU | |
| Shared Network | Architecture | Linear(8196→512)→ReLU | Linear(8196→512)→ReLU |
| Policy Head | Architecture | Linear(512→9) | Linear(512→256) → ReLU → Linear(256→3) |
| | Output | Discrete actions (9) | Continuous actions (steering, throttle, brake) |
| Value Head | Architecture | N/A | Linear(512→256) → ReLU → Linear(256→1) |
| Activation | Hidden | ReLU | |
| | Output | Linear | Policy: Tanh/Sigmoid, Value: Linear |
| Parameters | Total | 4,277,417 | 4,536,484 |

## B.2 ANALYSIS OF COMPUTATIONAL TIME (WALL-CLOCK TIME)

In this section, we evaluate the wall-clock time of MLES against baselines to assess its computational efficiency in discovering viable policies. All experiments are conducted on a single machine equipped with an Intel Core i9-13980HX processor, 32 GB of RAM, and an NVIDIA GeForce RTX 4060 GPU.

We measure efficiency using two primary metrics:

- **Total Time:** The full runtime of a method until a predefined budget of environment resets is exhausted. This metric reflects the overall computational cost for a comparable amount of exploration.

- **Time to Threshold:** The time elapsed from the start of a run until a method first discovers a policy meeting a predetermined performance threshold. This metric quantifies the speed at which

a method solves the task. The thresholds are a score of 1.00 for Lunar Lander and 95.0 for Car Racing.

The computational time for DQN and PPO primarily consists of GPU-accelerated neural network training and CPU-based environment rollouts. For MLES and EoH, the time is composed of: (1) *Sample time*, the latency for OpenAI API calls (using gpt-4o-mini), encompassing network latency and MLLM inference; (2) *Eval time*, the CPU time for policy evaluation in the environment; and (3) minor overhead for population management.

Table 9: Wall-Clock Time Comparison of MLES and Baselines

| Task | Method | Total Time | Time to Threshold | Remarks |
|---|---|---|---|---|
| Lunar Lander | DQN | 1h 55min | **0h 31min** | |
| | PPO | **1h 44min** | 1h 11min | |
| | EoH | 1h 49min | 0h 57min | Sample time:∼20s; Eval time: ∼4.5s |
| | MLES | 1h 50min | 0h 36min | Sample time:∼20s; Eval time: ∼4.5s |
| Car Racing | DQN | 41h 58min | — | |
| | PPO | 32h 37min | 12h 0min | |
| | EoH | 6h 47min | — | Sample time:∼20s; Eval time: ∼45s |
| | MLES | **6h 28min** | **3h 16min** | Sample time:∼20s; Eval time: ∼45s |

*Note: "—" indicates that the method failed to reach the performance threshold within the given time budget. The "Sample time" and "Eval time" in the Remarks column refer to the approximate durations for a single policy generation (API call) and its subsequent evaluation, respectively.*

Table 9 presents the wall-clock time comparison. The results demonstrate that MLES is highly competitive in computational efficiency, even outperforming traditional DRL baselines on key metrics. Several key observations and insights can be drawn from these results:

**1. Efficiency from inherent parallelism.** For the Car Racing task, both MLES and EoH exhibit significantly lower Total Time than the DRL methods, an advantage stemming from the EC framework's inherently parallel nature. In our implementation, policy sampling and evaluation for MLES and EoH are parallelized across eight processes. This architecture is highly effective for tasks with long evaluation times (approx. 45s per rollout in Car Racing), enabling concurrent evaluation of the population. Conversely, for Lunar Lander, where Eval time is much shorter (approx. 4.5s), the benefits of parallelization are less pronounced, and total times are comparable across all methods. This scalability demonstrates that the MLES framework is well-positioned to tackle more computationally intensive tasks without facing prohibitive runtime bottlenecks.

**2. Superior Time to Threshold.** MLES demonstrates exceptional performance on the Time to Threshold metric, which measures problem-solving speed. On Lunar Lander, MLES (36min) is competitive with the fastest baseline, DQN (31min), and substantially faster than PPO (1h 11min). This advantage is magnified in Car Racing, where MLES reaches the threshold in just 3h 16min, nearly four times faster than PPO (12h). This superior efficiency is attributable to two factors: (1) the rapid exploration rate enabled by the parallel EC framework, and (2) the enhanced search effectiveness from targeted policy improvements, which are guided by our visual feedback-driven behavior analysis.

**3. Negligible overhead from multimodal inputs.** A key finding is that introducing image-based behavioral evidence in MLES does not incur a significant latency penalty in API calls compared to the text-only EoH. The Sample time for both remained around 20 seconds. This demonstrates that the inference efficiency of current MLLMs like gpt-4o-mini is sufficient to support our framework without creating a bottleneck, rendering the rich guidance from multimodal feedback a computationally viable enhancement.

### B.3 INFLUENCE OF BEHAVIORAL EVIDENCE AND PROMPT ENGINEERING ON POLICY DISCOVERY

This section investigates the influence of different forms of BE and prompt engineering strategies on the policy discovery process. We use M1 as the test operator and construct various few-shot prompts that incorporate diverse forms of BE and specific instructions. Specifically, we evaluate the following five configurations:

- **M1**: The baseline operator derived from EoH, without any additional input.

- **M1_M**: Incorporates image inputs. The prompt explicitly instructs the MLLM to provide a detailed description and analysis of the image content.

- **M1_M$^\dagger$**: Includes image input, but without explicit instructions for description or analysis. The LLM directly optimizes the policy based on the provided image.

- **M1_T**: Introduces textual trajectory coordinates, as illustrated in Fig. 7(c). The prompt explicitly instructs the MLLM to describe and analyze this information in detail.

- **M1_M$^\ddagger$**: Employs a two-stage process. The image is first processed by an MLLM 1 to generate a detailed textual description, which is then passed to MLLM 2 for M1-level reasoning and policy generation. This configuration is akin to using an LLM that lacks direct access to the image.

To quantitatively assess the impact of these different BE forms and prompt instructions, we initialized our experiments with five distinct populations, each exhibiting significant evolutionary potential. These populations are carefully selected by MLES from an established policy evolution process for the Lunar Lander task. Specifically, we identify five generations that have previously demonstrated substantial progress and select their immediate preceding populations as our initial populations. This selection criterion ensures that our starting points possess considerable room for improvement, allowing for clearer observation of the operators' effects. Each M1 operator variant is applied to these populations for two generations. The entire experiment is repeated five times to ensure statistical robustness. We measure impact by observing both the average performance improvement across the population and the improvement in the best policy within each population.

Table 10: Average performance improvement in population and best policy across different configurations for the M1 operator.

|  | M1 | M1_M | M1_M$^\dagger$ | M1_T | M1_M$^\ddagger$ |
|---|---|---|---|---|---|
| Population improvement | +19.43% | +24.55% | +20.21% | +20.29% | +19.73% |
| Best policy improvement | +1.18% | +4.39% | +2.37% | +3.64% | +2.95% |

Table 10 summarizes the average improvements in population performance and best policy performance brought about by these operators. Several key observations and insights can be drawn from these results:

**1. Integrating BE significantly boosts policy discovery efficiency.** A foundational finding is that the incorporation of BE consistently leads to improved performance compared to the baseline M1 operator. The baseline M1, without any BE, achieves a population improvement of +19.43% and a best policy improvement of +1.18%. In contrast, all configurations that utilize BE (M1_M, M1_M$^\dagger$, M1_T, M1_M$^\ddagger$) exhibit higher gains in both population and best policy performance. For instance, M1_M demonstrates the highest population improvement of +24.55% and the most significant best policy improvement of +4.39%. This clearly indicates that providing MLLMs with relevant behavioral evidence significantly aids in fine-grained policy evaluation and enables more targeted policy optimization. The evidence acts as a crucial guide, steering the MLLM toward more effective policy spaces.

**2. The impact of interpretability and richness of BE.** Both M1_M and M1_T demonstrate substantial improvements in best policy performance, with +4.39% and +3.64%, respectively. While both forms of BE prove beneficial, the visual representation of the lander's behavior and posture in images is arguably more expressive, intuitive, and easier for the MLLM to interpret than raw textual state traces. This superior interpretability and rich expressiveness enable the MLLM to analyze policy behavioral patterns more deeply. These results support our claim, as discussed in Appendix D.1, that the most crucial aspect of BE is the quality and inherent interpretability of the information it conveys, along with how effectively this information can be leveraged by MLLMs for comprehensive behavioral analysis and subsequent policy refinement. The richer, more direct semantics embedded in visual data appear to offer a distinct advantage.

**3. The benefit of direct image input.** Comparing M1_M (direct image input) with M1_M$^\ddagger$ (two-stage processing where an image is first described by an MLLM into text, then passed to the policy generator), we observe a clear performance disparity. M1_M achieves a population improvement of +24.55% and a best policy improvement of +4.39%, whereas M1_M$^\ddagger$ yields slightly lower gains at +19.73% for population and +2.95% for best policy. When an image transforms into a textual description by another model, even if that description is "detailed," there is an inherent risk of information loss or misinterpretation. For an MLLM serving as a policy generator, directly accessing the raw image appears more advantageous than relying on a cascaded interpretation. This suggests that the rich, nuanced information present in the original image may be difficult to fully capture and convey through an intermediate textual representation, at least with current LLM capabilities. This outcome highlights the value of direct multimodal processing for inherently multimodal tasks, as it avoids potential bottlenecks and accumulated errors introduced by an intermediate textualization step.

**4. The advantage of explicit instructions for behavioral analysis.** The contrast between M1_M and M1_M$^\dagger$ strongly highlights the advantage of providing explicit instructions for behavior analysis within the prompt. M1_M consistently performs better in both population (+24.55%) and best policy (+4.39%) improvement compared to M1_M$^\dagger$ (+20.21% and +2.37% respectively). Both configurations receive image input, but M1_M explicitly instructs the LLM to provide a detailed description and analysis of the image, while M1_M$^\dagger$ does not. This gap emphasizes that while MLLMs can implicitly derive information from raw image input to aid inference, explicit directives to analyze BE can provide rich, relevant context. This promotes the activation of pertinent knowledge within the MLLM. By encouraging a "thinking aloud" process, such instructions enable the LLM to synthesize more comprehensive and actionable insights, which are subsequently utilized to generate more effective policy modifications. This finding is consistent with observations in Chain of Thought and similar works Yu et al. (2025); Wang & Venkatesh (2025).

In summary, these experiments underscore that both the form of BE and the strategy of prompt engineering are crucial factors in maximizing the effectiveness of MLES-driven policy discovery. A well-crafted context, combining expressive and interpretable BE with clearly instructed analytical processes during MLLM inference, significantly improves reasoning and enables more targeted and effective policy optimization. Our findings suggest that for complex tasks requiring a nuanced understanding of behavior, direct multimodal input, coupled with explicit analytical prompts, represents a superior approach.

### B.4 ABLATION STUDY

We conduct an ablation study to assess the individual contributions and the collective synergy of the key components within the MLES framework. This study evaluates variants of MLES where specific operators or mechanisms are removed. For clarity, we define the key variants as follows:

- MLES$^\dagger$ (w/o Exploration): This variant removes both exploration operators (E1 and E2). The policy search relies solely on refining existing policies based on multimodal feedback (M1_M and M2_M), without generating fundamentally new strategies or generalizing from multiple parents.
- MLES$^\ddagger$ (w/o Multimodal Modification): This variant removes both multimodal modification operators (M1_M and M2_M). The policy search proceeds using only the code and thoughts of parent policies (E1 and E2), foregoing any targeted refinement based on BE.
- w/o Thought: This variant removes the natural language "Thought" description from the prompt. The MLLMs must infer the policy's logic and intent solely from the raw code, without the high-level semantic guidance that explains the strategy.

We also evaluate the removal of each of the four operators individually, denoted by the prefix "w/o" (e.g., w/o E1). The average performance across three independent runs for each variant is summarized in Table 11. The metric $\Delta$Perf(%) quantifies the performance degradation relative to the full MLES framework. As shown in Table 11, each component contributes significantly to the overall performance of the MLES framework.

Table 11: Ablation study results on two control tasks.

| Method | Lunar Lander | | Car Racing | |
| --- | --- | --- | --- | --- |
| | Average | $\Delta$Perf (%) | Average | $\Delta$Perf (%) |
| MLES | 1.090 | 0.00% | 98.70 | 0.00% |
| w/o E1 | 1.093 | +0.17% | 87.53 | -11.32% |
| w/o E2 | 1.082 | -0.68% | 89.04 | -9.79% |
| MLES $^\dagger$ | 1.081 | -0.77% | 85.17 | -13.71% |
| w/o M1_M | 0.997 | -8.54% | 86.71 | -12.15% |
| w/o M2_M | 1.020 | -6.41% | 84.77 | -14.12% |
| MLES$^\ddagger$ | 0.629 | -42.24% | 83.16 | -15.75% |
| w/o Thought | 1.038 | -4.75% | 88.80 | -10.04% |

**The critical role of multimodal modification.** The most pronounced performance degradation is observed in the MLES$^\ddagger$ variant, where removing both M1_M and M2_M results in a catastrophic drop of 42.24% in Lunar Lander and a significant 15.75% decline in Car Racing. This finding strongly validates our core hypothesis: leveraging BE for targeted policy refinement is the primary driver of MLES's effectiveness. Without the ability to analyze behavioral shortcomings (M1_M) and fine-tune parameters (M2_M) based on concrete execution feedback, the policy search becomes vastly less efficient and is prone to stagnation.

**The importance of exploration.** The removal of exploration operators (MLES[†]) also leads to a notable performance decline, especially a 13.71% drop in the more complex Car Racing environment. This indicates that operators designed to synthesize fundamentally new strategies (E1) and generalize from successful existing ones (E2) are essential for navigating challenging policy spaces and escaping local optima. Interestingly, removing E1 in the simpler Lunar Lander task did not degrade performance but instead yielded a slight improvement (+0.17%). A plausible explanation is that for a less complex problem space, the aggressive exploration of E1 might introduce counterproductive variance. Given a fixed search budget, its absence allows the search to focus more effectively on refining already promising solutions. This strongly contrasts with its critical role in Car Racing, underscoring that E1's contribution is most significant in tasks demanding broad exploration.

**Synergistic effect.** While both operator types are important, the multimodal modification operators demonstrate a more foundational impact. However, the full MLES model outperforms all ablated variants, underscoring the synergistic relationship between exploration and modification. The exploration operators (E1, E2) act as the "engine of creativity," broadening the search by introducing diverse policy structures into the population. Subsequently, the multimodal modification operators (M1_M, M2_M) act as the "engine of refinement," grounding the search by improving these policies based on empirical evidence.

These results underscore the importance of both exploration operators and multimodal modification operators. By integrating these two aspects, MLES effectively balances a broad, creative search of the policy space with a deep, evidence-driven refinement of promising solutions. This powerful combination is key to its ability to efficiently discover high-performing policies.

**The role of Thought.** A design choice in MLES is to represent policies as a combination of executable code and a natural language "Thought" describing the underlying strategy. Removing this "Thought" leads to a substantial performance drop of 4.75% on Lunar Lander and 10.04% on Car Racing. This underscores the importance of semantic guidance, which helps MLLMs to quickly comprehend the policy and perform a more coherent search in the language space, thereby accelerating the policy discovery process. This observation aligns with prior findings in the EoH paper Liu et al. (2024a).

## B.5 Evaluations on Additional Control Tasks

The benchmarks employed in the main paper, Lunar Lander and Car Racing, are originally selected to span a broad spectrum of difficulty levels, ranging from **discrete action control** with **8-dimensional state inputs** to **continuous control** with **high-dimensional image observations**. This selection aims to maximize empirical diversity within a reasonable experimental scope.

To further assess the adaptability of MLES beyond these domains, we conduct additional experiments on the **Inverted Pendulum** task. We specifically design 10 challenging instances characterized by significant variations in initial angles and positions. The objective is to discover a single programmatic policy capable of balancing the pendulum for a full duration of 500 steps across all varying instances. The quantitative results are presented in Table 12.

Table 12: Performance comparison on the Inverted Pendulum task across 10 challenging instances. The maximum possible score (steps) is 500.

| Method | Avg. Steps |
|---|---|
| Initial Policy (PID baseline) | 327.9 |
| MLES with Initial Policy | **500.0** |
| MLES w/o Initial Policy | **500.0** |

As shown in Table 12, MLES successfully discovered a programmatic control policy that robustly handles all instance variations, achieving perfect scores regardless of policy initialization. We will continue the extension of MLES to diverse control tasks and update the open-source repository with these new environments.

## B.6 Generalization via Policy Ensemble

Finding a single policy that generalizes perfectly to all possible scenarios is often unattainable in practice. The performance degradation on test instances, observed across all methods in Table 1, illustrates this common generalization gap. This gap is not necessarily a flaw of any specific method, but rather an inherent characteristic of complex problems that demand case-specific solutions.

A significant, inherent advantage of MLES is that its evolutionary process produces not just a single policy, but a diverse population of high-performing candidate policies. We present performance heatmaps in Fig. 6,

which plot the scores of each policy in the final population across ten distinct test instances for two tasks. The varied color patterns illustrate that the discovered policies possess complementary strengths: some policies excel in particularly challenging instances where others struggle. This diversity confirms that MLES effectively explores and maintains a rich portfolio of distinct policies. Consequently, this presents a compelling opportunity to enhance generalization by applying ensemble decision-making to this set of high-performing, complementary policies.

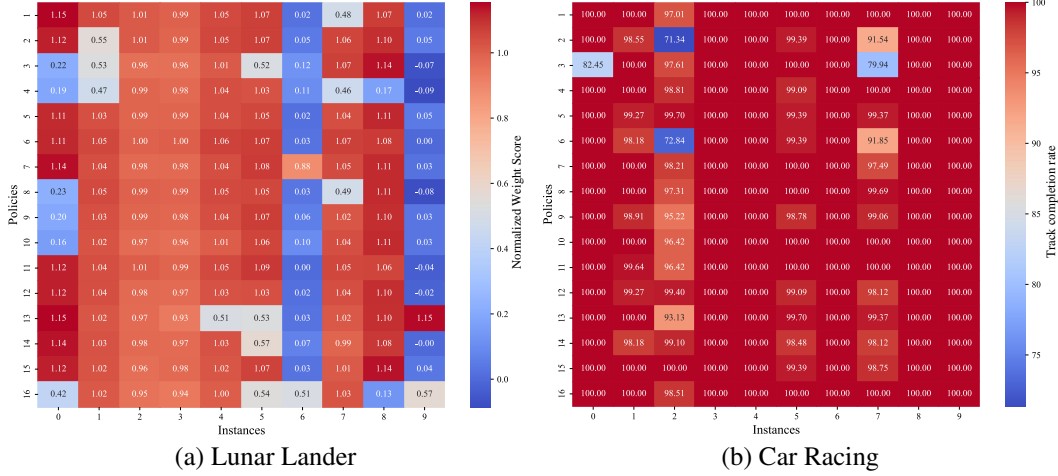

(a) Lunar Lander             (b) Car Racing

Figure 6: Performance heatmaps of the final MLES policy populations on (a) Lunar Lander and (b) Car Racing. Each row corresponds to one of the 16 final policies, and each column corresponds to one of 10 distinct test instances. The varied color patterns across rows for both tasks vividly illustrate the strategic diversity and complementary strengths within the discovered policy pools.

To validate this potential, we conduct an investigation using a straightforward ensembling strategy, with the results reported in Table 13. For the final policy population, the ensemble's action is selected by majority vote (for discrete tasks) or by averaging the outputs (for continuous tasks), and each policy is assigned an equal weight. MLES[†] and EoH[†] denote the ensemble versions of the respective methods.

Table 13: Effect of Policy Ensemble on LES Performance

| Method | Lunar Lander | | Car Racing | |
|---|---|---|---|---|
| | Train | Test | Train | Test |
| EoH | 1.053 | 0.776 | 89.808 | 79.286 |
| EoH[†] | 0.938↓ | 0.856↑ | 82.978↓ | 84.567↑ |
| MLES | 1.090 | 0.819 | 98.704 | 96.358 |
| MLES[†] | 1.032↓ | 0.901↑ | 96.904↓ | 97.921↑ |

The results are highly encouraging. Despite the simplicity of the ensemble method, it produced a significant improvement in test performance for both MLES and EoH. It is worth clarifying that a slight decline in training performance is also observed after ensembling. This outcome is a well-known characteristic of the bias-variance tradeoff, where ensembling acts as a form of regularization to improve generalization, sometimes at the cost of slightly increased bias on the training set.

Crucially, the complementarity of the policies in the population is achieved organically, without employing any explicit diversity-preservation techniques common in the EC field. This indicates substantial, yet-to-be-tapped potential to further improve MLES to address generalization issues. Looking forward, we believe this provides a promising direction for future work, which could focus on explicitly cultivating a "specialist team" of policies. This might involve exploring more advanced techniques, such as:

- **Feature-Aware Instance Clustering:** Automatically grouping similar problem instances based on their features to train or select specialized policies for each cluster.
- **Cooperative Co-evolution:** Designing efficient evolutionary mechanisms that encourage the policy population to collectively cover the full spectrum of training instances, rewarding policies for succeeding where others fail.

- **Adaptive Collaborative Decision-Making:** Creating dynamic ensembling mechanisms where policies "vote" on the final action with varying confidence levels based on the current state.

Overall, while not the primary focus of this paper, MLES demonstrates inherent properties and substantial potential that could be harnessed to tackle the challenge of generalization.

# C DETAILS OF QUANTITATIVE METRICS IN MLES

## C.1 QUANTITATIVE METRICS FOR POLICY EVALUATION IN MLES

In MLES, quantitative metrics serve as indicators of policy performance, analogous to the episode reward function in DRL. These metrics provide a consistent basis for ranking and selecting candidates during the policy evolution process. Unlike DRL, where each action is evaluated with immediate rewards in an online fashion, MLES evaluates the overall performance of a policy across a set of instances. In other words, MLES utilizes sparse rewards to guide policy learning, focusing on the final performance outcome.

This approach offers a practical advantage in that it eliminates the need for frequent manual design of intermediate rewards. We only need to focus on transforming the ultimate task goal into an evaluable metric, which can then support automated policy discovery. This not only simplifies the process but also alleviates the computational burden associated with reward design. However, this comes with the challenge of lacking direct evaluation of intermediate behaviors, which may potentially lead to issues such as reward hacking. To address this limitation, MLES incorporates *behavioral evidence* (BE), which supplements the evaluation of intermediate actions. A more detailed discussion of this can be found in Section **2.3** and Appendix **D**.

Given a set of training instances, MLES aims to maximize (or minimize) the quantitative metric across the policies in its policy pool. During the evaluation process, the performance of a policy is typically assessed in parallel across all relevant instances, and the overall quantitative metric is computed as a weighted average of the metrics for each instance. Through iterative optimization, this process generates policies that are well-suited to instances drawn from the same distribution as the training instances.

## C.2 QUANTITATIVE METRICS FOR BENCHMARK TASKS

This subsection outlines the quantitative metrics employed for evaluating policies in the Lunar Lander and Car Racing tasks.

**Lunar Lander**  The quantitative metric for evaluating performance in the Lunar Lander task is the Normalized Weight Score (NWS), which is defined as follows:

$$\text{NWS} = \frac{R}{200} \times 0.6 + (1 - \min(\frac{C}{100}, 1)) \times 0.2 + S \times 0.2 \tag{2}$$

where $R$ is the mean episode reward, $C$ is the mean fuel consumption, and $S$ is the success rate of completing the landing task, all computed over multiple instances.

For each training instance, the episode reward is calculated by summing the rewards across all steps taken by the policy during execution. The reward for each step is determined by the following criteria:

- Reward increases/decreases based on the proximity to the landing pad.
- Reward increases/decreases based on the lander's speed (slower movement is rewarded).
- Reward decreases based on the tilt of the lander (less tilt is better).
- An additional reward of 10 points is awarded for each leg in contact with the ground.
- Penalties are imposed for using side or main engines during flight (-0.03 and -0.3 points per frame, respectively).
- A penalty of -100 points is incurred for crashing, and a bonus of +100 points is awarded for a successful landing.

A policy is considered successful if it achieves an episode reward of at least 200, which contributes to the first term in the NWS calculation. The policy's success is also determined by its ability to land safely, which is reflected in the third term of the NWS. A policy with an NWS greater than 1 indicates a 100% safe landing rate, with larger values indicating more fuel-efficient landings. To ensure a fair comparison, DRL algorithms also use the default reward function described above.

**Car Racing**    Regarding the Car Racing task, the Gym environment provides a reward of -0.1 points for each frame, with an additional reward of +1000/N for each track tile visited, where $N$ is the total number of tiles in the track. This reward function incentivizes agents to complete the race track efficiently while avoiding driving off the course.

For the Car Racing task, we intuitively use the track completion rate as the quantitative metric for each instance. MLES aims to maximize the mean track completion rate across the training instances as its objective for policy discovery. In our experiment, DRL adopts the default reward function from the Gym environment, as described above.

# D    THE FORMATS AND NECESSITY OF BEHAVIOR EVIDENCE

This section provides a detailed discussion of the potential formats of Behavior Evidence (BE) and the construction techniques involved, along with the specific forms adopted in this work and the rationale behind our choices.

## D.1    FORMATS OF BEHAVIOR EVIDENCE

The primary objective of BE is to represent the behavior patterns associated with a given policy, thereby helping MLLMs identify policy shortcomings and facilitate subsequent improvements. BE serves as a crucial supplement to the evaluation of intermediate actions, addressing the inherent limitations of purely quantitative metrics such as episode reward and success rate, which often fail to provide granular insights into policy execution.

The precise format of BE is secondary to the quality and interpretability of the information it conveys, and how effectively this information can be leveraged by MLLMs. MLLM training paradigms, relying on human-provided images and corresponding natural language descriptions, progressively align model capabilities with human understanding. As evidenced by related work, current MLLMs demonstrate a level of visual comprehension closely mirroring human perception, enabling them to interpret information in a semantically meaningful way.

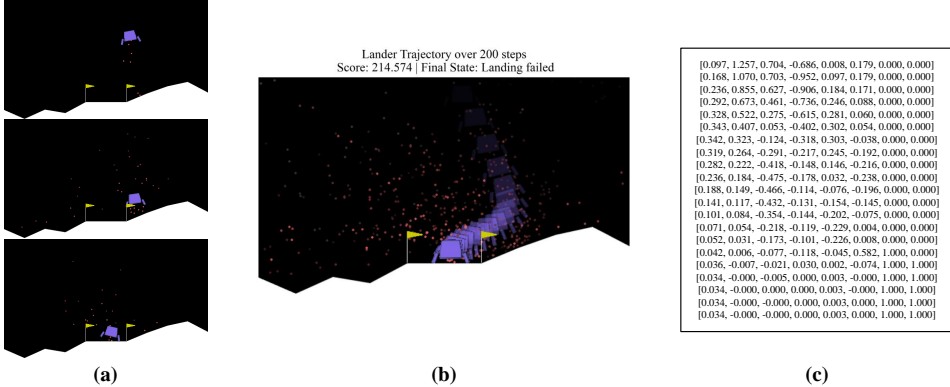

Figure 7: Examples of BE in the Lunar Lander task. (a) Trajectory videos visualizing the agent's motion over time. (b) Pose overlays summarizing sequential states in a single image. (c) State traces presenting a symbolic representation of the agent's behavior.

Fig. 7 illustrates three potential BE formats within the context of the Lunar Lander task: videos, images, and textual state traces. While these formats convey the same underlying information, their interpretability and computational overhead for MLLM input vary significantly. Videos offer a complete visualization of the entire process but incur substantial computational expense. With appropriate preprocessing, images can effectively convey the control process while maintaining relatively low computational costs. Conversely, textual descriptions, though offering the lowest computational cost, are less intuitive for both human and MLLM interpretation, demanding greater cognitive effort for comprehension.

It is important to note that agent behaviors in numerous tasks are challenging to represent clearly using purely textual formats. For instance, in the Car Racing task within our benchmarks, coordinate-based trajectories or performance records in textual form fail to convey the intricate and subtle interactions between the car and the track. Such representations do not capture crucial nuances of how the agent controls its motion and the corresponding dynamic outcomes, which are vital for a comprehensive policy analysis. In contrast, visual representations offer significantly richer expressiveness for these dynamics, thereby facilitating better policy understanding by humans. As previously discussed, MLLMs can also leverage visual data to interpret and

analyze the policy's control process, thereby underpinning the visual feedback-driven behavior analysis within the MLES.

Furthermore, real-world control tasks, such as bipedal robot walking, drone stabilization, or autonomous driving, frequently involve complex dynamics that are inherently difficult to articulate or fully capture through textual means. Visual formats are therefore superior for conveying these complex behaviors. In our experiments, to strike a balance between comprehensibility for both humans and MLLMs and computational efficiency, we opt to employ images as the primary format for BE. The specific methodologies for constructing them are detailed in the subsequent section.

## D.2 BE FOR TWO BENCHMARKS

This section details the construction of BE for Lunar Lander and Car Racing tasks, addressing their unique characteristics.

**Lunar Lander.** The Lunar Lander task features a static background, with only the lander object in motion. Given this setup, we can effectively capture the entire behavioral pattern by using a frame stacking technique with transparency. As illustrated in Fig. 8, this process involves selecting frames at fixed intervals and applying a transparency effect to each one before stacking them. The resulting BE is a single image that visualizes the lander's pose and trajectory at key moments, effectively compressing the time-series information of an entire episode into a compact, visual representation.

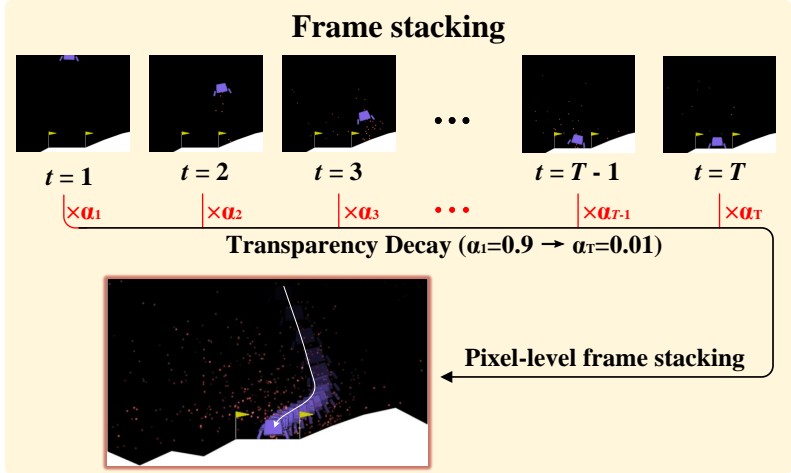

Figure 8: Frame stacking for BE construction for the Lunar Lander task

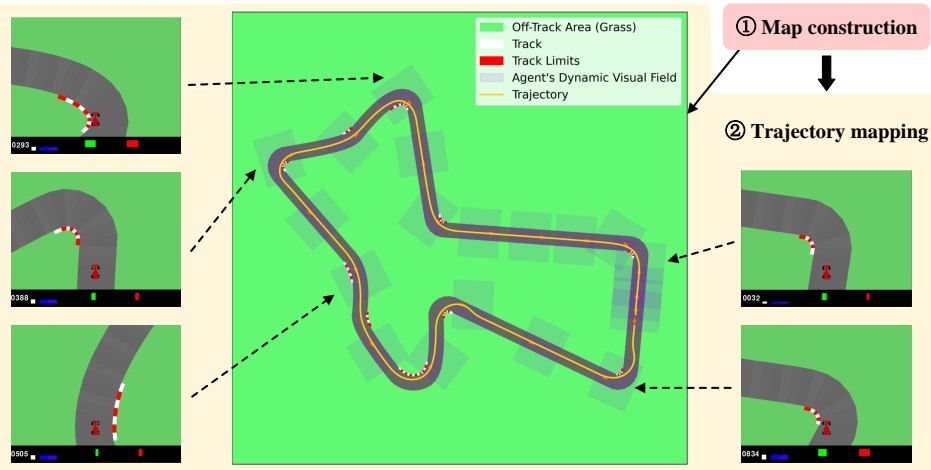

Figure 9: Trajectory mapping for BE construction for the Car racing task

**Car Racing.** The Car Racing task is a top-down environment where the camera continuously follows the car, and the background is dynamic. For this reason, the frame stacking method is not applicable. Instead, we propose a BE construction method based on a global map and trajectory mapping, as shown in Fig. 9. The process begins by obtaining the complete coordinates of the track from the environment's backend to construct a global track map. The car's movement trajectory is then recorded throughout the episode and plotted onto this map upon completion. However, a simple trajectory alone is insufficient for explaining the policy's decision-making process.

To address this limitation, we annotate the agent's dynamic visual field at fixed environment step intervals along the trajectory. This enhancement provides two key benefits: (1) The density of the visual fields directly reflects the car's speed: a denser distribution indicates a slower speed, and vice versa. (2) These visual fields provide crucial context, enabling MLLM to analyze the specific conditions that led to suboptimal actions, such as veering off the track. This targeted contextual information facilitates more precise reasoning and informed policy improvements.

### D.3 NECESSITY OF BE

This section presents a qualitative analysis of BE's necessity, highlighting how it addresses a key limitation of quantitative metrics: their inability to evaluate the intermediate actions.

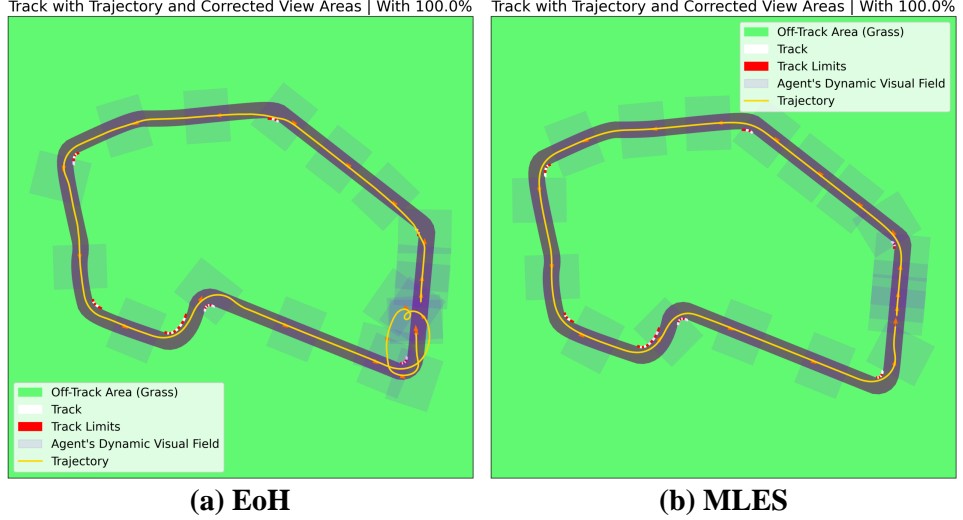

(a) EoH                                       (b) MLES

Figure 10: Trajectories of policies generated by EoH and MLES on a specific track, both achieving a perfect score

Fig. 10 illustrates the performance of policies designed by EoH and MLES on the same Car Racing track. Both policies achieve a perfect score, completing 100% of the track. However, a closer inspection of their trajectories reveals a critical difference. The EoH policy's trajectory deviates significantly from the track on the final turn, only completing the course by executing a large corrective maneuver. This behavior is a classic example of "reward hacking," a common issue in RL where agents exploit flawed reward functions to achieve a high score through suboptimal or unintended behaviors Skalse et al. (2022). In contrast, the MLES policy's trajectory is both rational and efficient, closely mirroring a human driver's racing line.

Because EoH's evolutionary process is driven solely by a quantitative performance metric, it fails to evaluate the quality of the policy's intermediate actions and thus cannot identify and correct such superficially high-performing but ultimately flawed policies. This leads to two significant problems: (1) Deceptively high-scoring policies can persist within the population, hindering the overall evolutionary progress. (2) Policies may achieve strong results on training instances but exhibit poor generalization to new, unseen instances.

In our approach, MLES utilizes BE as a crucial supplement to quantitative metrics, allowing MLLMs to perform in-depth behavioral analysis. This qualitative input enables the system to identify and proactively rectify these "fake-excellent" policies. Furthermore, for sound but underperforming policies, BE enables the MLLM to provide targeted feedback on specific improvements, thereby accelerating the policy discovery process. This highlights the indispensable role of BE in the LES paradigm for achieving robust and generalizable policy learning.

# E PROMPTS

## E.1 TEMPLATES FOR CONSTRUCTING PROMPTS FOR SPECIFIC OPERATORS

This section details the prompt templates utilized for policy generation within the MLES framework. Figs. 11, 12, 13, and 14 illustrate the specific prompt templates for operators E1, E2, M1_M, and M2_M, respectively. In all presented templates, black text indicates fixed components, red placeholders denote task-specific elements, and blue placeholders represent variables that evolve throughout the evolutionary process.

You are assigned as an expert to participate in the following task: **"[Task Description Placeholder]"**
I have **"[Value Placeholder]"** existing algorithms with their codes as follows:
The No. 1 algorithm and the corresponding code are:
**"[Thought Placeholder]"**
**"[Code Placeholder]"**
The No. 2 algorithm and the corresponding code are:
**"[Thought Placeholder]"**
**"[Code Placeholder]"**
...
Please help me create a new algorithm that has a totally different form from the given ones.
1. First, describe your new algorithm and main steps in one sentence. The description must be inside within boxed {}.
2. Next, implement the following Python function:
**"[Code Template Placeholder]"**

Figure 11: Prompt template for the E1 operator.

You are assigned as an expert to participate in the following task: **"[Task Description Placeholder]"**
I have **"[Value Placeholder]"** existing algorithms with their codes as follows:
The No. 1 algorithm and the corresponding code are:
**"[Thought Placeholder]"**
**"[Code Placeholder]"**
The No. 2 algorithm and the corresponding code are:
**"[Thought Placeholder]"**
**"[Code Placeholder]"**
...
Please help me create a new algorithm that has a totally different form from the given ones but can be motivated from them.
1. Firstly, identify the common backbone idea in the provided algorithms.
2. Secondly, based on the backbone idea describe your new algorithm in one sentence. The description must be inside within boxed {}.
3. Thirdly, implement the following Python function:
**"[Code Template Placeholder]"**

Figure 12: Prompt template for the E2 operator.

## E.2 TASK-SPECIFIC PROMPTS USED IN EXPERIMENTS

This section demonstrates the design of task descriptions and code templates for applying MLES, using two control problems from our experiments as examples. Briefly, the task description communicates the requirements for the control policy to be designed, while the code template defines the programmatic policy's code interface that MLES needs to generate. Figs. 15 and 16 present the task descriptions provided to MLES for the Lunar Lander and Car Racing tasks, respectively. Figs. 19 and 20 show the corresponding code templates.

**Task description.** The task description serves as a problem statement for MLES, akin to an assignment given by a teacher to a student. First, we establish the overall context of the policy discovery task by clearly stating the

You are assigned as an expert to participate in the following task: **"[Task Description Placeholder]"**
We have a working algorithm that needs optimization. Below are its concept, implementation, and execution results:
Concept: **"[Thought Placeholder]"**
Implementation: **"[Code Placeholder]"**
Execution results visualization for the algorithm: **"[Behavioral Evidence Placeholder]"**
Please start by providing a detailed description and analysis of the execution result, enclosed within single quotes (' '). Next, based on your analysis, optimize the algorithm by following these steps:
1. Analyze why the results were produced in relation to the algorithm. Identify its weaknesses and areas for improvement, and enclose your analysis within square brackets [ ].
2. Propose an enhanced algorithm. Use concise language to describe the core idea of your algorithm, and enclose the core idea within curly braces {}.
3. Implement the enhanced algorithm using the following Python function template:
**"[Code Template Placeholder]"**

Figure 13: Prompt template for the M1_M operator.

You are assigned as an expert to participate in the following task: **"[Task Description Placeholder]"**
We have a working algorithm that needs optimization. Below are its concept, implementation, and execution results:
Concept: **"[Thought Placeholder]"**
Implementation: **"[Code Placeholder]"**
Execution results visualization for the algorithm: **"[Behavioral Evidence Placeholder]"**
Please start by providing a detailed description and analysis of the execution result, enclosed within single quotes (' '). Next, based on your analysis, optimize the algorithm by following these steps:
1. Parameter Analysis:
- Identify all key parameters and their functions.
- Determine which parameters should be modified to improve results.
- Explain why these specific changes would help.
- All content related to Parameter Analysis must be enclosed within brackets [ ].
2. Create a new algorithm that has a different parameter settings of the algorithm provided. Use concise language to describe the core idea of your algorithm, and enclose the core idea within curly braces {}.
3. Implement the enhanced algorithm using the following Python function template:
**"[Code Template Placeholder]"**

Figure 14: Prompt template for the M2_M operator.

objective: what control policy MLES needs to design. This primes the MLLMs to inherently prepare relevant knowledge for policy generation. Second, we explicitly articulate the desired characteristics of an optimal policy, guiding the LLM's value alignment with human intent. Furthermore, we can provide a more detailed description of the control task to facilitate precise programming by MLES. For example, in Fig. 16, we elaborate on observation details, such as the type of observation data and key visual elements. More comprehensive information tends to activate more relevant knowledge within the LLM.

> Implement a novel heuristic strategy function that guides the lander in selecting actions step-by-step to achieve a safe landing. At each step, an appropriate action could be chosen based on the lander's current state and previous state, with the objective of reaching the target location in as few steps as possible. A 'safe landing' is defined as a touchdown with low vertical speed, upright orientation, and both angular velocity and angle close to zero, and both legs in contact with the ground.

Figure 15: Task description for the Lunar Lander task.

> Write a Python function that serves as a control strategy for an agent in a top-down car racing environment.
> **Environment Overview**
> In this environment, the agent is required to drive a car along a race track. The primary objective is to cover as much of the track surface as possible before the time limit expires. To accomplish this, the agent needs to efficiently navigate the track by controlling the car's steering, throttle, and brake.
> **Observation Details**
> The agent's observation consists of a $96 \times 96 \times 3$ RGB image representing the top-down view of the environment. The following key visual elements can be identified in this image:
> - Car: Red (approximately $[\approx 202, <10, <10]$).
> - Track: Gray (approximately $[\approx 102, \approx 102, \approx 102]$).
> - Off-track grass: Greenish (approximately $[\approx 102, \approx 204, \approx 102]$).
> - Curbs (Sharp Turns): High-contrast red and white (approximately $[>240, <20, <20]$ and $[>240, >240, >240]$).
> **Inputs at Each Time Step**
> At every time step, the agent receives the following information:
> - The current RGB observation of the environment.
> - The current speed of the car.
> - The previous RGB observation and the previous action taken by the agent.
> **Function Requirements**
> The Python function should incorporate a control policy that combines visual perception from the RGB observations and past information (previous observation and action). This policy should enable the agent to maintain optimal control of the car, keep the car on the track, and maximize the efficiency of lap completion.

Figure 16: Task description for the Car Racing task.

**Code Template.**    The code template defines the communication protocol and can optionally furnish MLES with expert knowledge. The specific templates used in our experiments are illustrated in Fig. 17 and 18. A code template typically consists of three parts: function declaration (function name and parameters), docstring, and function body.

Within the function declaration and docstring, we must pre-define the parameters that the policy and environment will exchange, ensuring compatibility with the environment's interface. This involves: (1) identifying the parameters the environment or system provide to the policy and defining them as input variables in the function declaration; (2) detailing the type, dimension, range, and meaning of these input parameters in the docstring; and (3) clearly specifying the policy's output in the docstring, typically the actions the agent can take. Additionally, we can incorporate prior knowledge and hints within the docstring to enhance the effectiveness of policies generated by MLES. As shown in Fig. 18, we can include "Notes" sections within the docstring to help MLES better understand the environment and offer specific guidance. For the function body, we can simply provide a return statement to ensure function completeness, or we can additionally include existing expert heuristics.

```python
import numpy as np

def choose_action(s: list, last_action: int, s_pre: list) -> int:
    """
    Selects an action for the Lunar Lander to achieve a safe landing at the target location
        (0, 0).
    Args:
        s (list or np.ndarray): The current state of the lander. Elements:
            s[0] - horizontal position (x)
            s[1] - vertical position (y)
            s[2] - horizontal velocity (v_x)
            s[3] - vertical velocity (v_y)
            s[4] - angle (radians)
            s[5] - angular velocity
            s[6] - 1 if the first leg is in contact with the ground, else 0
            s[7] - 1 if the second leg is in contact with the ground, else 0
        last_action (int): The action taken in the previous step. One of:
            0 - do nothing
            1 - fire left orientation engine
            2 - fire main (upward) engine
            3 - fire right orientation engine
        s_pre (list or np.ndarray): The state of the lander *before* the last action was
             executed.
    Returns:
        int: The chosen action for the next step. One of:
            0 - do nothing
            1 - fire left orientation engine
            2 - fire main (upward) engine
            3 - fire right orientation engine
    """
    a = np.random.choice([0, 1, 2, 3])
    return a
```

Figure 17: Code template for the Lunar Lander Task

```python
import numpy as np
import cv2
def choose_action(observation, car_speed, pre_action, pre_observation):
    """
    Determine the next action for the Car Racing agent.
    This function takes into account the current state (observation and speed), the previous
        action, and the previous observation.
    Notes:
    - The car in this environment is a powerful rear-wheel-drive vehicle. Avoid accelerating
        while turning sharply,
      as this can easily lead to loss of control.
    - Occasionally, track segments (e.g., after a U-turn) may appear in the observation but
        are not part of the immediate drivable path. These should be distinguished to avoid
        premature or incorrect decisions.
    - Avoid coming to a complete stop, as this may prevent the car from finishing the race.
    Args:
        observation (np.ndarray): The current state observed by the agent, represented as a 96
            x96 RGB image of the car and race track from a top-down view (shape: (96, 96, 3))
            .
        car_speed (float): The current speed of the car.
        pre_action (np.ndarray): The action taken by the car in the previous step, represented
             as a 3-element array.
        pre_observation (np.ndarray): The observation received when the previous action was
            taken. It has the same shape and format as `observation` (i.e., a 96x96 RGB image
            ).
    Returns:
        np.ndarray: The action selected by the agent for the next step, represented as an
            array of shape (3,) where:
                    - Index 0: Steering, where -1 is full left, +1 is full right (range: [-1,
                         1]).
                    - Index 1: Gas, (range: [0, 1]).
                    - Index 2: Braking, (range: [0, 1]).
    """
    action = np.array([0.0, 0.0, 0.0])
    return action
```

Figure 18: Code template for the Car Racing Task

```python
import numpy as np

def choose_action(s: list, last_action: int, s_pre: list) -> int:
    angle_targ = s[0] * 0.5 + s[2] * 1.0  # angle should point towards center
    if angle_targ > 0.4:
        angle_targ = 0.4  # more than 0.4 radians (22 degrees) is bad
    if angle_targ < -0.4:
        angle_targ = -0.4
    hover_targ = 0.55 * np.abs(
        s[0]
    )  # target y should be proportional to horizontal offset

    angle_todo = (angle_targ - s[4]) * 0.5 - (s[5]) * 1.0
    hover_todo = (hover_targ - s[1]) * 0.5 - (s[3]) * 0.5

    if s[6] or s[7]:  # legs have contact
        angle_todo = 0
        hover_todo = (
            -(s[3]) * 0.5
        )  # override to reduce fall speed, that's all we need after contact

    a = 0
    if hover_todo > np.abs(angle_todo) and hover_todo > 0.05:
        a = 2
    elif angle_todo < -0.05:
        a = 3
    elif angle_todo > +0.05:
        a = 1
    return a
```

Figure 19: The provided initial policy for the Lunar Lander task.

```python
import numpy as np
import cv2
def choose_action(observation, car_speed, pre_action, pre_observation):
    action = np.array([0.0, 0.0, 0.0])
    # Gray track detection parameters (RGB 95-115 range with +-5% tolerance)
    gray_low = 95
    gray_high = 115

    # Create 3D gray detection mask (all RGB channels within range)
    gray_mask = (
            (observation[:, :, 0] >= gray_low) & (observation[:, :, 0] <= gray_high) &
            (observation[:, :, 1] >= gray_low) & (observation[:, :, 1] <= gray_high) &
            (observation[:, :, 2] >= gray_low) & (observation[:, :, 2] <= gray_high)
    )

    gray_indices = np.argwhere(gray_mask)
    center_x = np.mean(gray_indices[:, 1]) if len(gray_indices) > 0 else observation.shape[1]
        // 2
    car_position = observation.shape[1] // 2
    offset = center_x - car_position

    steering_angle = np.clip(offset / 100.0, -1.0, 1.0)
    action[0] = steering_angle

    if abs(offset) > 10:
        action[1] = 0.0
        action[2] = 0.2
    else:
        action[1] = 0.8
        action[2] = 0.0

    gray_density = np.sum(gray_mask) / (observation.shape[0] * observation.shape[1])
    if gray_density < 0.1:
        action[1] = 0.4
        action[2] = 0.3
    return action
```

Figure 20: The provided initial policy for the Car Racing Task

## F  GENERATED PROGRAMMATIC POLICY

This section presents the best policies discovered by MLES, including the code, thought, and performance on a range of instances. By comparing the final MLES-generated policies with the initial policies used to seed the population (see Fig. 19 and 20), we observe significant upgrades in both feature processing and control logic.

**Lunar Lander.**  Fig. 21 shows the code for the best policy discovered for the Lunar Lander task. The thought of this policy provided by MLES is: "The core idea of the new algorithm is to further increase the sensitivity to vertical velocity, making the lander more responsive to abrupt changes, while also lowering the action thresholds to enable quicker decisions for stability control." Fig.22 displays the performance of this policy across several instances, highlighting its ability to achieve safe landings consistently.

```python
def choose_action(s: list, last_action: int, s_pre: list) -> int:
    """
    Selects an action for the Lunar Lander to achieve a safe landing at the target location
        (0, 0).
    Args:
        s (list or np.ndarray): The current state of the lander. Elements:
            s[0] - horizontal position (x)
            s[1] - vertical position (y)
            s[2] - horizontal velocity (v_x)
            s[3] - vertical velocity (v_y)
            s[4] - angle (radians)
            s[5] - angular velocity
            s[6] - 1 if the first leg is in contact with the ground, else 0
            s[7] - 1 if the second leg is in contact with the ground, else 0
        last_action (int): The action taken in the previous step. One of:
            0 - do nothing
            1 - fire left orientation engine
            2 - fire main (upward) engine
            3 - fire right orientation engine
        s_pre (list or np.ndarray): The state of the lander *before* the last action was
            executed.
    Returns:
        int: The chosen action for the next step. One of:
            0 - do nothing
            1 - fire left orientation engine
            2 - fire main (upward) engine
            3 - fire right orientation engine
    """
    angle_targ = (s[0] * 0.5 + s[2] * 0.5)  # Designed for better orientation control
    angle_targ = np.clip(angle_targ, -0.6, 0.6)  # Slightly wider bounds for orientation

    # Update hover target with amplified sensitivity to vertical speed
    hover_targ = 0.3 * np.abs(s[0]) + 0.7 * (s[3] ** 2)  # Greater weight for hover
        stabilizing

    # Calculate actions for adjustments
    angle_todo = (angle_targ - s[4]) * 1.6 - (s[5]) * 1.2  # More aggressive response for
        angle adjustments
    hover_todo = (hover_targ - s[1]) * 1.3 - (s[3]) * 0.4  # Enhanced weight for vertical
        control

    # Stabilizing when legs are in contact
    if s[6] or s[7]:
        angle_todo = 0  # Maintain upright position priority
        hover_todo -= (s[3]) * 0.4  # Stronger adjustment for vertical speed reduction

    # Decision-making based on refined thresholds and more responsive measures
    a = 0
    if hover_todo > np.abs(angle_todo) and hover_todo > 0.10:  # Slightly tighter hover
        threshold
        a = 2  # Fire main engine
    elif angle_todo < -0.15:  # Increased sensitivity for left engine
        a = 3
    elif angle_todo > 0.15:  # Increased sensitivity for right engine
        a = 1
    return a
```

Figure 21: Code of the best discovered policy for the Lunar Lander task.

**Car Racing.**  Fig. 23 shows the code for the best policy discovered for the Car Racing task. The core idea of this policy is to dynamically adjust the steering sensitivity and throttle response based on the car's speed

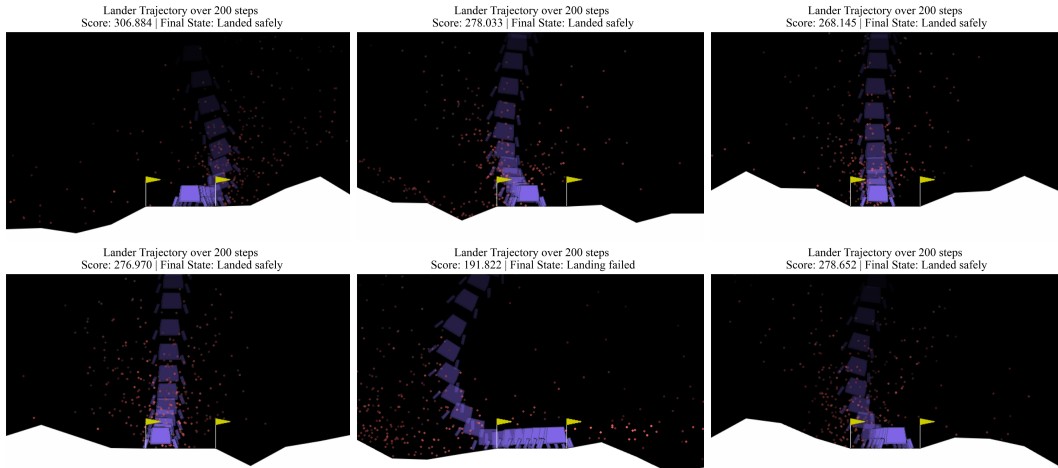

Figure 22: Performance of the best discovered policy on different instances in the Lunar Lander task. It shows the lander's successful landings and stability under various conditions.

and previous actions. This adjustment enables smoother control over the car's acceleration and deceleration, ensuring optimal performance during races. Furthermore, the policy incorporates an improved method for assessing track boundaries, focusing on smoother transitions between acceleration and deceleration. The policy intelligently uses the car's speed to modulate steering and throttle, allowing for better control in turns while avoiding overly sharp maneuvers that could destabilize the vehicle. Fig. 24 illustrates the policy's performance on several instances, demonstrating its ability to complete the race smoothly, with racing lines that resemble those of human drivers.

## G  DETAILS OF INVERTED PENDULUM EXPERIMENTS

In this section, we provide full transparency regarding the Inverted Pendulum experiments. To ensure reproducibility and facilitate inspection of the automated discovery process, we present the task description prompt, the code template, the initial policy used for the "With Initial Policy" setting, and the final best-performing programmatic policy discovered by MLES.

The task description provided to the LLM is shown in Fig. 25. The code template used for the "From Scratch" setting is displayed in Fig. 26. The initial heuristic policy provided for the "With Initial Policy" setting is shown in Fig. 27. Finally, the best-performing programmatic policy discovered by MLES is presented in Fig. 28. As observed in the code, the policy generated by MLES exhibits significant improvements in logical complexity, robustness, and handling of edge cases compared to the initial heuristic, demonstrating the effectiveness of the evolutionary refinement process.

## H  THE USE OF LARGE LANGUAGE MODELS

LLMs are utilized in two primary ways in this study: first, to serve as programmatic policy generators within the MLES framework; second, to help refine the manuscript by enhancing the clarity of the language, while the authors maintain full responsibility for the content and direction.

```python
def choose_action(observation, car_speed, pre_action, pre_observation):
    """
    Determine the next action for the Car Racing agent.
    Args:
        observation (np.ndarray): The current state observed by the agent, represented as a 96
            x96 RGB image of the car and race track from a top-down view (shape: (96, 96, 3))
            .
        car_speed (float): The current speed of the car.
        pre_action (np.ndarray): Action taken by the agent in the previous step, represented
            as a 3-element array.
        pre_observation (np.ndarray): The observation received when the previous action was
            taken. It has the same shape and format as `observation` (i.e., a 96x96 RGB image
            ).
    Returns:
        np.ndarray: The action selected by the agent for the next step, represented as an
            array of shape (3,) where:
                    - Index 0: Steering, where -1 is full left, +1 is full right (range: [-1,
                        1]).
                    - Index 1: Gas, (range: [0, 1]).
                    - Index 2: Braking, (range: [0, 1]).
    """
    hsv = cv2.cvtColor(observation, cv2.COLOR_RGB2HSV)
    # Define masks for track, off-track, and curbs detection
    track_mask = cv2.inRange(hsv, (0, 0, 60), (179, 255, 200))  # Track pixels
    off_track_mask = cv2.inRange(hsv, (40, 100, 100), (90, 255, 255))  # Off-track pixels
    curbs_mask = cv2.inRange(hsv, (180, 100, 100), (255, 255, 255))  # Curbs detection

    # Analyze the center region for better decision making
    height, width = track_mask.shape
    center_region = track_mask[int(height / 3):int(2 * height / 3), int(width / 4):int(3 *
        width / 4)]
    # Initialize the action array
    action = np.zeros(3)
    # Identify whether track is present and where
    track_detected = np.sum(center_region) > 0
    if not track_detected:
        action[2] = 1.0  # Full brake if no track detected
    else:
        # Identify track pixel positions
        track_pixels = np.column_stack(np.where(center_region > 0))
        mean_x = np.mean(track_pixels[:, 1]) if len(track_pixels) > 0 else center_region.shape
            [1] / 2

        # Steering adjustment with consideration for immediate curvature
        steering = (mean_x - (center_region.shape[1] / 2)) / (center_region.shape[1] / 2)
        action[0] = np.clip(steering, -1, 1)  # Normalize steering value

    # Adaptive throttle management based on speed
    if car_speed < 1.0:
        action[1] = 1.0  # Full throttle if the car is very slow
    else:
        # Adjust throttle considering steering
        if abs(action[0]) > 0.5:
            action[1] = max(0.0, 1.0 - abs(action[0]))  # Reduce throttle during significant
                turns
        else:
            action[1] = min(1.0, action[1] + 0.05)  # Gradual increase on straights

    # Enhanced braking logic considering sharp turns
    if abs(action[0]) > 0.6 and car_speed > 2.0:
        action[2] = min(0.4, action[2] + 0.1)  # Moderate braking for sudden turns
    else:
        action[2] = 0.0  # No brake needed on straights

    # Prevent the car from stalling
    if np.all(pre_action == [0, 0, 1]) and car_speed < 0.5:
        action[1] = max(action[1], 0.5)  # Ensure some throttle to avoid stalling
    return action
```

Figure 23: Code of the best discovered policy for the Car Racing task.

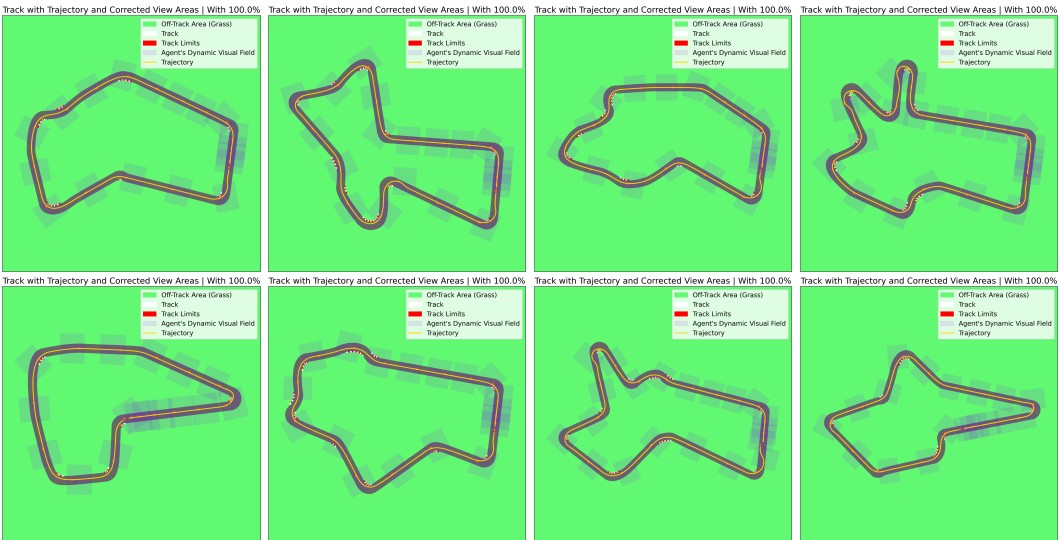

Figure 24: Performance of the best discovered policy on different instances in the Car Racing task. These figures show the car's smooth track completion and race line strategy, similar to human racing behavior.

Implement a novel heuristic strategy function that guides the agent (the cart) in selecting actions step-by-step to keep the pole balanced. At each step, an appropriate action should be chosen based on the agent's current state and previous state, with the objective of maximizing the number of steps the pole remains balanced. A "successful balance" is defined as keeping the pole upright without falling over within the environment's predefined limits (pole angle within approx. $\pm 0.209 \text{ rad}$ ($\pm 12°$) and cart position within approx. $\pm 2.4$ units).

Figure 25: Task description for the Inverted Pendulum task.

```python
import numpy as np

def choose_action(s: list, last_action: int, s_pre: list) -> int:
    """
    Selects an action for the CartPole to keep it balanced.

    Args:
        s (list or np.ndarray): The current state of the CartPole. Elements:
            s[0] - Cart Position
            s[1] - Cart Velocity
            s[2] - Pole Angle (radians)
            s[3] - Pole Angular Velocity

        last_action (int): The action taken in the previous step. One of:
            0 - push cart to the left
            1 - push cart to the right

        s_pre (list or np.ndarray): The state of the CartPole *before* the last action was
             executed. Elements:
            s_pre[0] - Cart Position (previous state)
            s_pre[1] - Cart Velocity (previous state)
            s_pre[2] - Pole Angle (previous state)
            s_pre[3] - Pole Angular Velocity (previous state)

    Returns:
        int: The chosen action for the next step. One of:
            0 - push cart to the left
            1 - push cart to the right
    """
    a = np.random.choice([0, 1])
    return action
```

Figure 26: Code template for the Inverted Pendulum Task

```python
import numpy as np

def choose_action(s: list, last_action: int, s_pre: list) -> int:
    pole_angle = s[2]
    pole_velocity = s[3]

    angle_weight = 4.0
    velocity_weight = 2.0

    control_signal = (pole_angle * angle_weight) + (pole_velocity * velocity_weight)

    if control_signal > 0:
        action = 1
    else:
        action = 0
    return action
```

Figure 27: The provided initial policy for the Inverted Pendulum task.

```python
def choose_action(s: list, last_action: int, s_pre: list) -> int:
    """
    Select an action for the CartPole to keep it balanced.
    """

    # Unpack state variables
    x = float(s[0])          # cart position
    x_dot = float(s[1])      # cart velocity
    theta = float(s[2])      # pole angle
    theta_dot = float(s[3])  # pole angular velocity

    # --- Heuristic parameters ---
    dt = 0.02                    # short prediction horizon (typical CartPole timestep)
    k_theta = 60.0               # proportional gain on pole angle
    k_theta_dot = 3.5            # derivative gain on pole angular velocity
    k_x = 1.0                    # correction on cart position
    k_x_dot = 0.8                # damping on cart velocity
    k_inertia = 0.6              # bias to avoid rapid action flipping

    # Predict short-term pole angle to detect accelerating fall (Euler)
    theta_pred = theta + theta_dot * dt

    # If angle is accelerating outward from upright -> increase angular velocity weight
    theta_dot_factor = 1.8 if abs(theta_pred) > abs(theta) else 1.0

    # Inertia term: last action push direction
    inertia = 1.0 if last_action == 1 else -1.0

    # Control signal:
    #   > 0 : push right
    #   < 0 : push left
    control = (
        k_theta * theta
        + k_theta_dot * theta_dot * theta_dot_factor
        + k_x * x
        + k_x_dot * x_dot
        + k_inertia * inertia
    )

    # --- Safety overrides for imminent failures ---

    # Strong correction if pole is tilted and still accelerating outward
    angle_limit = 0.15  # radians (~8.6 degrees)
    if abs(theta) > angle_limit and abs(theta_dot) > 0.4:
        direction = 1.0 if theta * theta_dot >= 0 else -1.0
        control += (k_theta * 0.5) * direction

    # Prevent cart drifting near the position boundary
    position_limit = 1.7  # buffer before ~+-2.4 environment limit
    if abs(x) > position_limit:
        direction = 1.0 if x > 0 else -1.0
        control += (-k_x * 3.0) * direction

    # Action selection
    return 1 if control > 0 else 0
```

Figure 28: Code of the best discovered policy for the Inverted Pendulum task.

