# OpenReview forum: "Multimodal LLM-assisted Evolutionary Search for Programmatic Control Policies"
_ICLR.cc/2026/Conference — ICLR 2026 Poster_

### Official Review · Reviewer_EJft · 2025-10-22

**Soundness:** 3
**Presentation:** 3
**Contribution:** 3
**Rating:** 6
**Confidence:** 4

**Summary:**

The paper presents Multimodal LLM Assisted Evolutionary Search (MLES) as an approach to automatically discovering control policies (as opposed to conventional deep reinforcement learning or genetic algorithms). MLES involves using multimodal LLMs to generate policies, which are then evaluated and summarized by a multimodal LLM to produce summary feedback, which is then provided to the policy generator to help cover behavior gaps that are not adequately reflected in the final score. In the midst of this loop, LLMs serve to orchestrate which policies are chosen to be evaluated/refined, LLMs summarize the feedback, and LLMs generate prompts for LLMs to write new policies. In particular, the authors claim that adding behavioral evidence (summaries of behavior that are not necessarily quantitative) is a key enabler of their method.

The MLES system is compared to a prior LLM + Evolutionary Search algorithm, to DQN, and to PPO on the Lunar Lander and Car-Racing gym environments. The scope of experiments demonstrates success on structured and unstructured state spaces, as well as continuous and discrete actions. The results are strong, showing that the MLES system solves tasks as effectively as conventional RL, sometimes more effectively, while maintaining a smaller standard error and yielding a finished policy that a compute scientist can inspect and potentially interpret.

The appendix is full of useful information, supplementary results, and further ablations of the method. It is clear that significantly care was taken to thoroughly examine the MLES system, and the authors have carefully explained the contributions from each individual component of the method (for example, studying the effects of different exploration operators and multimodal-modification operators). Finished policies (as code) are also included in the supplement.

**Strengths:**

+ The paper is very clearly explained and the MLES method is generally easy to understand. While there are certain elements that make more sense in theory than in practice, the overall design of the system is clear and the paper is likely reproducible.
+ The method is benchmarked very thoroughly on two domains, LunarLander and CarRacing, with ablations of different components, comparisons to more conventional deep RL, and comparisons to prior LLM + evolutionary algorithms work.
+ The understandability of the method is evaluated by real humans, and the final policies generalize well to new test instances of the same task. These are both very strong results and indicate the promise of the method.

**Weaknesses:**

- Novelty: While the paper is interesting and the results are strong, the central ideas are not significantly different from prior work [1], down to the figure similarity. The central innovation in this work is the multimodality and the expression of “behavioral evidence”, although this is somewhat buried in the presentation of the entire system as a novel contribution. In reality, most of the system is ported from prior work, and the core novelty and contribution here seems to be on the VLM descriptions of failures and qualitative feedback to update the model’s heuristics.
- Generalizability: MLES shows strong results on train and test environments, but it is not clear how effectively this method would generalize without (1) strong scaffolding code or heuristics, and (2) structured state data. While the CarRacing experiment does use unstructured data, the method also seems to rely on having semantic state information (such as speed) injected into the policy.
- Exploration diversity: MLES makes a few design decisions that seem to bias it very strongly towards iterative refinement of a single strong heuristic, rather than a more conventional exploration into potentially highly negative rewards. While this may be desirable for safety-critical applications, it may be undesirable for discovering new behaviors.

[1] Fei Liu, Tong Xialiang, Mingxuan Yuan, Xi Lin, Fu Luo, Zhenkun Wang, Zhichao Lu, and Qingfu Zhang. Evolution of heuristics: Towards efficient automatic algorithm design using large language model. In Forty-first International Conference on Machine Learning, 2024a.

**Questions:**

* How does parent selection affect the method? Currently, parents are sampled according to their quantitative results via a method that biases very heavily towards the high-performers. Conventionally, genetic algorithms might consider a much broader sample of parents (such as the top 50%). Does a wider sampling method derail the approach? In other words, is there really only 1 viable strategy that MLES just iteratively refining?
* How does E2 (the “generalize across both parent control policies” prompt) handle control policies that cannot be reconciled? If parent A suggests going left around an obstacle, and parent B suggests going right, there is no blending of these two policies.
* Behavioral evidence is necessary, but policies are ranked on quantitative metrics. Is there no way to incorporate behavioral feedback into the ranking?
* Is the compute comparison with PPO and DQN fair, given that you get 10K * 16 (rollouts) for the MLES method?
* Low standard error suggests that there may not be much variability in the policies that are being tested or explored. Is this the case? Is the low standard error a reflection of the fact that MLES effectively iterates on the same couple of heuristics?
* How well can MLES recover from a bad local optimum? For example, if the Lunar Lander was seeded with 1 heuristic that stays aloft the entire episode and 1 heuristic that attempts to land but crashes, it seems possible that the system would iteratively refine the "stay aloft" policy indefinitely, perhaps never learning to land because of the strong negative quantitative penalty to failing a landing (therefore leading that attempted landing policy to not be sampled for iterative refinement).

---

> ### Author Response · Authors · 2025-11-19
> **Response to Reviewer EJft (1/3)**
>
> We sincerely thank the reviewer for the detailed assessment and constructive feedback. We are encouraged by the recognition of our extensive analysis and the potential of the MLES framework. Below, we address each concern and question in detail.
>
> ### **W1: Clarifying Novelty and Improving Presentation**
>
> We appreciate the reviewer’s insight that our original presentation may obscure the novelty of our framework. While MLES shares the evolutionary backbone with EoH, our work fundamentally diverges in how search is driven and guided. EoH relies on scalar feedback and natural-language "thoughts" to elicit LLM reasoning. In contrast, MLES introduces visual feedback to transition **from blind, scalar-driven search to targeted, rationale-driven refinement**.
>
> Our choice to **adapt the EoH with minimal structural changes was a deliberate methodological choice** for a controlled ablation study. By keeping the evolutionary framework consistent, we isolated "visual feedback" as the critical variable, scientifically proving that multimodal insight is the key driver of performance in control tasks, rather than complex new genetic operators or different search strategies.
>
> However, it seems to bury our contribution inside a system that looks like EoH. To make this clearer, we have revised Section 2 to:
> 1. Explicitly highlight MLES as a general visual-feedback-enhanced automated discovery framework, not an incremental extension of EoH.
> 2. Clarify that EoH serves only as a reference implementation to illustrate how multimodal feedback can strengthen an existing evolutionary-search pipeline.
> 3. Clearly state our intention for making only minimal modifications to EoH.
>
> We respectfully invite the reviewer to examine the updated revision and would be happy to further improve the clarity of the presentation based on additional suggestions.
>
> ### **W2: Concerns on Generalizability**
>
> To address concerns about reliance on scaffolding code, we conducted additional experiments where MLES starts ***entirely from scratch*** (no initial policy or heuristics) with 2000 LLM queries:
>
> |                                   | Lunar Lander | Car Racing |
> | :-------------------------------- | :-----------: | :--------: |
> | MLES (with Initial Policy, GPT-4o-mini) |     1.090     |   98.07    |
> | MLES (w/o Initial Policy, GPT-4o-mini)  |     0.667     |   91.96    |
> | MLES (w/o Initial Policy, GPT-5-mini)   |     1.013     |   100.0    |
>
> Results show that while starting from scratch requires more exploration, **MLES is fully capable of discovering high-performance policies without scaffolding code or heuristics**. Notably, when powered by the more capable GPT-5-mini, MLES achieves excellent results even without an initial policy. We would like to emphasize that the ability to ingest initial policies is an **optional advantage to accelerate search**, not a prerequisite.
>
> Regarding the observation on CarRacing (unstructured image + structured state): We argue this is a strength of MLES. MLES acts like a human programmer, autonomously leveraging all available information to solve the task. Moreover, the visual-processing portion (Fig. 21) plays a dominant role in the final decision logic, illustrating that MLES effectively handles unstructured data.
>
> We hope this addresses concerns about generalizability.
>
> ### **W3：Exploration diversity**
>
> We would like to clarify that **MLES does not merely refine a single strong heuristic.** Our pipeline integrates exploration operators (E1, E2), targeted-modification operators (M1_M, M2_M), and probability-based parent selection, which together prevent the search process from over-allocating computational resources to any single policy.
>
> Figure 5 provides concrete evidence of this. The semi-transparent yellow lineage path connecting the ancestors of the final policy shows that MLES does not simply evolve one dominant individual throughout the run. A closer inspection further reveals that Ind 198 and Ind 770 were generated by E2 and E1, introducing entirely new behavioral patterns into the population. This demonstrates that MLES maintains meaningful exploration during the run.

---

> > ### Author Response · Authors · 2025-11-19
> > **Response to Reviewer EJft (2/3)**
> >
> > ### **Q1： How does parent selection affect the method?**
> >
> > Parent selection is a core component of the MLES framework. As in traditional Evolutionary Computation, the choice of the selection mechanism dictates the trade-off between exploration and exploitation. A wider sampling method would increase exploration but dilute selection pressure. As established in [1], insufficient selection pressure can significantly slow down convergence. Given that MLES operates with LLM calls (unlike traditional EC with cheap offspring generation), maintaining sufficient selection pressure is critical for computational efficiency.
> >
> > Therefore, we employ *exponential rank selection*. It favors high performers to drive convergence **but does not** preclude the emergence of novel policies via exploration operators (E1/E2). This ensures we do not strictly iterate on a single "best" policy. Together with the evidence discussed in **W3**, this confirms that MLES does not collapse into refining a single policy.
> >
> > [1] T. Blickle and L. Thiele, "A Comparison of Selection Schemes Used in Evolutionary Algorithms," in Evolutionary Computation, vol. 4, no. 4, pp. 361-394, Dec. 1996
> >
> > ### **Q2：How does E2 reconcile conflicting policies (e.g., Left vs. Right)?**
> >
> > The LLM resolves such conflicts through semantic synthesis rather than naive token-level merging. Unlike traditional genetic crossover, which might blend code syntax, the LLM operates at the logic level.
> >
> > **Empirically, we observed that** when parents exhibit opposing strategies (e.g., Parent A steers left, Parent B steers right), the LLM does not average the commands. Instead, it synthesizes a generalized control logic, such as "dynamically choose direction based on the obstacle's relative position." The conflict is resolved by elevating the policy to a higher level of reasoning that accommodates both scenarios conditionally.
> >
> > ### **Q3：Can behavioral feedback be incorporated into ranking?**
> >
> > Our current design employs a specialized division of labor to ensure the effectiveness of the policy discovery process:
> > - Quantitative Metrics: Used for selection (ranking), serving as the ground-truth objective function to drive convergence.
> > - Behavioral Evidence: Used for policy generation, acting as the diagnostic signal to guide ***how to improve***.
> >
> > We appreciate the reviewer's suggestion to incorporate behavioral feedback into ranking. While theoretically attractive, we identified three practical challenges during the design process:
> >
> > 1. **Lack of an objective behavior scoring function**: Behavioral feedback requires semantic understanding and normative judgment (e.g., defining "smoothness" or "cautiousness"). Implementing a learned or LLM-based scorer would introduce significant complexity and computational cost.
> >
> > 2. **Difficulty in combining heterogeneous metrics**: Behavioral scores and objective values operate on different scales and semantics. Combining them into a scalar for ranking introduces hyperparameter sensitivity regarding how to weigh these distinct signals.
> >
> > 3. **Redundancy with the reward function**: Empirically, poorly performing policies almost always correspond to undesirable behaviors. Thus, the objective reward already captures a significant portion of the signal needed for ranking/selection.
> >
> > We view this integration as a promising direction for future work. However, our core claim remains unaffected: visual behavioral evidence effectively guides rationale-driven policy refinement and contributes to measurable performance gains. We welcome further discussion on this topic.

---

> > > ### Author Response · Authors · 2025-11-19
> > > **Response to Reviewer EJft (3/3)**
> > >
> > > ### **Q4： Is the compute comparison fair?**
> > >
> > > We would like to clarify a misunderstanding regarding the computational budget. The "10K" refers to the total number of rollouts (environment resets), which is kept consistent across all methods to ensure fairness.
> > >
> > > Breakdown for Lunar Lander:
> > > - MLES/EoH: 2000 LLM queries $\times$ 5 rollouts/query = 10,000 rollouts. (Note: The factor of 5 arises because each generated policy is evaluated on 5 training instances, consuming 5 rollouts per query.)
> > > - Baselines (PPO/DQN): Trained for the equivalent number of Total rollouts.
> > >
> > > Thus, the comparison is strictly fair in terms of environmental interaction cost.
> > >
> > > ### **Q5： Low standard error vs. policy variability**
> > >
> > > We wish to clarify that the low standard error in Figures 3 & 4 reflects the robustness of MLES across independent runs (i.e., the algorithm consistently finds high-quality policy regardless of the random seed), not a lack of diversity within a single run's population.
> > >
> > > Evidence: As illustrated by the grey dots in Figure 5, the fitness values within a single population exhibit significant spread during the search process. This demonstrates that MLES maintains sufficient diversity to explore the policy space effectively before converging to an optimal policy.
> > >
> > > We have further addressed the mechanisms for maintaining diversity in our responses to W3 and Q1.
> > >
> > > ### **Q6： Ability to Recover from Bad Local Optima**
> > >
> > > MLES includes two dedicated exploration operators (E1, E2), enabling escape from local optima. Moreover, the LLM-based generator can recognize that a “hovering” heuristic cannot achieve the landing goal, and therefore produces corrective strategies.
> > >
> > > An Evidence: Ind 198 in Fig. 5 exemplifies a successful recovery from a sub-optimal policy via the E2 operator and later becomes part of the optimal. This illustrates MLES’s ability to recover from poor initial strategies.
> > >
> > > ---
> > >
> > > We thank the reviewer again for the insightful and professional comments. We will incorporate these clarifications and additional experimental evidence into the updated manuscript (A version with Section 2 reorganized has been submitted). We are happy to further discuss any remaining concerns.

---

> > > > ### Comment · Reviewer_EJft · 2025-11-25
> > > >
> > > > Thank you for your updates and your responses to my questions. I maintain that there may be generalization issues, in particular because the code policies seem to rely on CV2 and hand-crafted rules for image input (and similarly, may require heuristics and hacks for other modalities such as point clouds or audio). However, the additional experiments with no policy initialization and with GPT 5.1 are compelling, and address my concern about over-optimization for a single warm-started policy.
> > > >
> > > > With respect to the new draft, I appreciate the content updates and the refined statement of novelty. The first figure is quite unclear, however, and requires more explanation either in the main body or in the caption to explain what the reader is seeing, and how it explains the overall paper. As-is, that figure cannot really standalone, and the caption does not satisfactorily explain how it is drawing a distinction between the current paper and the prior work.

---

> ### Author Response · Authors · 2025-11-26
> **Response to Reviewer EJft (Follow-up)**
>
> Thank you for your continued feedback. We are encouraged that the additional experiments have alleviated your concerns. We appreciate your engagement with our work and have carefully addressed your remaining concerns below.
>
> ### **Q1: On Generalization and Tool Use**
>
> We appreciate this thoughtful comment regarding generalization. We want to clarify that the use of tools (like CV2)  is an **emergent behavior of the evolutionary process**, rather than a hard-coded dependency. We address this from three perspectives:
>
> **(1) Tools are selected by evolution, not hard-coded.**
>
> Our logs show that during the search, MLES autonomously explored multiple perception strategies, ranging from raw pixel manipulation (such as RGB thresholding, grayscale conversion, edge detection) to library calls. The cv2 component **was not forced upon the agent**; rather, it was introduced by the MLLM as a candidate strategy and remained in the final policy only because **it survived repeated selection pressure**. In other words, the evolutionary process identified cv2 as a robust and efficient tool for this specific task (especially in the image processing part) among competing strategies.
>
> **Evidence**: In the Car Racing task, even when we removed 'import cv2' in the ***from-scratch*** setting, MLES autonomously generated the import statements and eventually converged to a cv2-based policy (achieving a score of 100). This demonstrates MLES’s ability to actively discover and select the best tools from its knowledge base to solve the problem.
>
> **(2) "Tool Use" aligns with human engineering practices, enhancing Generalizability.**
>
> We respectfully argue that ***leveraging robust, widely-validated libraries enhances, rather than limits, generalizability***. Just as a human engineer would reasonably utilize **‘Librosa’ for audio** or **‘Open3D’ for point clouds**, MLES acts as an agent capable of synthesizing the appropriate Tool-Use Logic based on the input modality. This is not a "hack" but a higher-level cognitive capability. This avoids "reinventing the wheel" and significantly improves MLES's ability to handle unstructured data by using robust processing tools verified by the human community.
>
> **(3) Modularity supports scaling to complex scenarios.**
>
> As discussed in Appendix A.1 (Lines 767-780), the policies generated by MLES are inherently modular: perception logic is decoupled from control logic. This allows the input processing module to be easily upgraded. For scenarios where heuristic rules or standard libraries are insufficient, MLES can be directed to incorporate advanced foundation models (e.g., ViT or YOLO) as part of its perception module. This modularity ensures MLES remains adaptable to complex environments where simple heuristics fall short.
>
> In summary, the presence of cv2 in the final policy is a testament to MLES's capability to find good practical engineering solutions, confirming its potential to generalize effectively across different domains via intelligent tool use.
>
> ### **Q2: Clarifying Figure 1 and Distinction from Prior Work**
>
> Thank you for pointing out the ambiguity in Figure 1. We have revised the figure and its caption to ensure it is self-contained.
>
> **Clarification on the Figure's Intent:** The primary purpose of Figure 1 is to highlight a core contribution of MLES through a clear and impactful contrast with DRL (the dominant paradigm for generating control policies). Specifically, the figure illustrates that MLES, like DRL, directly interacts with the environment to discover high-performing policies, but unlike DRL, it preserves full transparency. We have updated the figure’s annotations to more clearly convey this point.
>
> Regarding **the distinction from prior LES methods** such as *Eureka* or *EoH*, we have ensured this comparison is explicitly addressed in the main body **immediately following Figure 1 (Lines 75–83)**：
> - Unlike Eureka, which focuses on shaping reward functions for DRL, MLES directly generates programmatic policies.
> - Unlike EoH, which is an automated discovery framework that relies solely on scalar metrics, MLES introduces visual feedback–driven behavior analysis to make the evolutionary process more transparent, traceable, and efficient.
>
> **Design Note:** We initially considered including DRL, LES, and MLES together in a single figure to emphasize their differences. However, illustrating three distinct paradigms created excessive visual clutter and reduced interpretability. Therefore, we chose to focus Figure 1 on the high-level contrast with DRL to capture the reader's attention, while providing the detailed distinctions from prior methods (like EoH) in the adjacent text.
>
> ---
>
> Thank you again for your constructive comments. The revised Figure 1 has been updated in the latest submission, and we sincerely invite you to examine it. We would be very happy to further discuss any remaining concerns and continue improving the quality of the paper.

---

> > ### Comment · Reviewer_EJft · 2025-11-26
> >
> > Thank you for the prompt response and for the figure caption revision. The revised caption and figure are a bit clearer.
> >
> > I appreciate the points raised on generalizability through emergent tool-use and library imports. My concerns have been addressed by the rebuttal, and I have accordingly raised my score.

---

> > > ### Author Response · Authors · 2025-11-28
> > >
> > > We sincerely appreciate your constructive suggestions, time, and efforts, as well as your decision to raise the score to 8 (Accept). Your feedback has significantly helped us improve the quality of our paper.

---

### Official Review · Reviewer_qyrE · 2025-10-27

**Soundness:** 3
**Presentation:** 3
**Contribution:** 2
**Rating:** 4
**Confidence:** 4

**Summary:**

This paper proposes the MLES (Multimodal LLM-assisted Evolutionary Search) framework, which uses Multimodal Large Language Models (MLLMs) combined with Evolutionary Computation (EC) to automatically discover programmatic control policies. Unlike the "black-box" policies of traditional DRL, MLES generates interpretable Python code. Its core innovation is integrating multimodal Behavioral Evidence (BE), especially visual feedback (e.g., images, trajectory maps), into the evolutionary loop. MLLMs analyze this BE alongside quantitative metrics (like reward scores) to identify failure patterns and guide policy improvements . Policies are represented as a combination of executable Code and natural language "Thought". Experiments on Lunar Lander and Car Racing show MLES generates transparent policies with performance comparable to the strong DRL baseline, PPO and DQN.

**Strengths:**

1. Figure 5 is nice, and it would to nice have more like this in the appendix. Papers discussing programmatic policies usually only present they final intrepretable policies, but the procedure of discovering is missing. I am happy to see such a procedure with the help pf evoluationary search.
2. Ablation study is provided, this is nice. This is especially helpful for a work which invoke MLLM with different prompts (E1, E2, M1_M, M2_M).
3. The final policies, as well as the searching process are intepretable, yet the performance is comparable with agents trained in DQN or PPO.

**Weaknesses:**

1. Usually this kind of framework is hard to generalize to other domains of tasks. This typically involve composing template carefully. The initial code template is good, but this also inject to much inductive bias.
2. The framework is highly dependent on MLLM's capabilities. Did you perform any ablation study that how the choices and settings of MLLMs could affect the overall performance?
3. Lunar Lander and Car Racing are important RL tasks, yet they are too simple. Did you ever apply this framework on other complex long-horizon RL tasks, even with image observation spaces? It would be very interesting if this work can be applied to neurosymbolic settings (i.e., utilizing higher-level programs to control lower-level policies). Such as the environment discussed in [1] and many others.

[1] Qiu and Zhu, Programmatic Reinforcement Learning without Oracles, ICLR 2022

**Questions:**

1. See weaknesses 2.
2. See weaknesses 3.
3. MLLM API call could be expensive, could you please give an simple analysis on how many tokens are used in each training, and what are the prices?
4. Can you elaborate how to constrcut multimodal few-shot prompt?
5. Can you elaborate how to acquire behaviroal evidence? How many frames of images do you need?
6. Can you give an example of "thought"? It would be nice to see the evolve of "thoughts" along with the training. "Thoughts" may change.

---

> ### Author Response · Authors · 2025-11-17
> **Response to Reviewer qyrE (1/3)**
>
> Thank you for your thoughtful and constructive feedback. We address each concern below and will incorporate additional clarifications and experiments in the revised manuscript.
>
> ### **W1: Concerns on Generalization and Inductive Bias of Code Templates**
>
> We respectfully clarify that **MLES does not rely on domain heuristics embedded in the code templates**. As stated in Lines 1447–1457, templates only define the communication protocol (i.e., function signatures). In all experiments, the function bodies would be set as empty or trivial (like: return 0), containing no heuristic behavior.
>
> In the original appendix, Fig. 17-18 showed the initial policies but were labeled as templates, which may have misled readers. In the revised version:
> - Fig. 17-18 will show the actual code templates,
> - The initial policies will be shown separately.
>
> To empirically address the concern about dependence on an initial policy (seed), we conducted additional experiments starting from scratch (without any initial policy) over 2000 LLM calls:
>
> |                                   | Lunar Lander | Car Racing |
> | :-------------------------------- | :-----------: | :--------: |
> | MLES (with Initial Policy, GPT-4o-mini) |     1.090     |   98.07    |
> | MLES (w/o Initial Policy, GPT-4o-mini)  |     0.667     |   91.96    |
> | MLES (w/o Initial Policy, GPT-5-mini)   |     1.013     |   100.0    |
>
> Results show that while starting from scratch requires more exploration, **MLES is fully capable of discovering high-performance policies without seeds**. Notably, when powered by the more capable GPT-5-mini, MLES achieves excellent results even without an initial policy. The ability to ingest an initial policy is an **optional feature** to accelerate search, **not a strict requirement**. This demonstrates that MLES is a general framework not bound by specific inductive biases. We hope this clarification addresses the reviewer's concerns on generalization and inductive bias.
>
>
> ### **W2 & Q1: Dependence on MLLM capabilities (Additional ablation study)**
>
> As we discuss in Appendix A.3, MLES's performance is naturally coupled with the underlying MLLM's capabilities. To quantify this, we compared GPT-5-mini (released Aug 2025) against GPT-4o-mini (released June 2024) under the same experiment setting.
>
> |      | Lunar Lander  |  | Car Racing | |
> |---------|:----------------------------:|:--------------:|:---------------------------:|:--------------:|
> |           | Score at 2000 samples        | Samples to reach 4o-mini's 2000-sample score | Score at 2000 samples        | Samples to reach 4o-mini's 2000-sample score |
> | MLES with Initial Policy (GPT-4o-mini) | 1.09                         | -              | 98.07                        | -              |
> | MLES with Initial Policy (GPT-5-mini)  | 1.065                        | 2000           | 100                          | 346            |
> | MLES w/o Initial Policy (GPT-4o-mini)  | 0.667                        | -              | 91.96                        | -              |
> | MLES w/o Initial Policy (GPT-5-mini)   | 1.013                        | 258            | 100                          | 209            |
>
>
> The results confirm that stronger MLLMs significantly boost MLES’s search efficiency (reducing sample complexity by ~5-10x in some cases) and final performance. This suggests that MLES benefits directly from the rapid advancement of foundation models, ensuring its longevity and scalability.

---

> > ### Author Response · Authors · 2025-11-17
> > **Response to Reviewer qyrE (2/3)**
> >
> > ### **W3 & Q2: Scalability to more complex tasks like Neurosymbolic Settings**
> >
> > We appreciate the reviewer’s insightful suggestion connecting MLES to neurosymbolic settings and the work of Qiu and Zhu [1]. We strongly agree that MLES is inherently well-suited for such tasks. Below, we provide a theoretical analysis to clarify this connection.
> >
> > The method π-RPL [1] designs specific DSLs (Affine, Ensemble, PID policies) and applies continuous relaxation to make the DSL differentiable for gradient-based optimization. Once trained, the higher-level programs can call and coordinate lower-level controllers to accomplish the control task.
> >
> > **MLES can, in principle, achieve the same objective.** For example, MLES can be used to evolve high-level programs that learn to combine and invoke existing low-level policies (e.g., PID controllers or other hand-crafted modules). If we treat a library of fixed “lower-level policies” as the available actions within the control task, then "discovering the high-level program" becomes structurally similar to the “policy design” problem studied in our Lunar Lander and Car Racing experiments. As demonstrated in our paper, MLES is capable of discovering such programmatic policies effectively.
> >
> > What is even more exciting is that **MLES avoids intrinsic constraints of π-RPL.** Unlike π-RPL, which requires carefully defining DSLs, specifying its grammar, and relaxing it for differentiability, MLES is free from these restrictions. MLES can directly generate any necessary Python function through interaction with the environment, offering substantially higher flexibility and theoretical scalability. In the future, MLES can also be extended to more sophisticated hierarchical design settings, where both high-level program structures and low-level controllers' parameters could be co-designed (an idea conceptually related to the system design perspective explored in [2]).
> >
> > Overall, we appreciate the reviewer’s insightful comments. We agree that neurosymbolic settings represent an exciting future direction for MLES, and we will incorporate this discussion into the revised manuscript. We also plan to explore this direction in our future work.
> >
> > **Clarification on the two used Benchmarks**: Lunar Lander and Car Racing were originally chosen to cover a broad range of difficulty (from discrete action to continuous control, and from 8-dimensional states to **high-dimensional image observations**) to maximize coverage within a reasonable experimental scope. We will continue extending MLES to additional tasks and will update the open-source repository accordingly, as this is crucial for broader community engagement with MLES.
> >
> > [1] Qiu and Zhu, Programmatic Reinforcement Learning without Oracles, ICLR 2022
> > [2] van Stein, Niki, Diederick Vermetten, and Thomas Bäck. "In-the-loop hyper-parameter optimization for llm-based automated design of heuristics." ACM Transactions on Evolutionary Learning (2024).

---

> > > ### Author Response · Authors · 2025-11-17
> > > **Response to Reviewer qyrE (3/3)**
> > >
> > > ### **Q3: MLLM API Cost Analysis**
> > >
> > > We provide a detailed cost analysis in Appendix A.2. While API calls incur costs, they are modest compared to the value of interpretability. For example, solving the Car Racing benchmark (approx. 2000 requests with GPT-4o-mini) cost **only ~$0.42 USD in total**.
> > >
> > > - About Token Usage: Text-only operators (E1, E2) use **~2k tokens/call**. Multimodal operators (M1, M2) use **~15k tokens/call** (including image tokens). Given that MLES produces transparent, white-box code, we believe this cost is highly justified.
> > >
> > > Encouragingly, this cost is expected to continue decreasing as MLLMs further advance in the future.
> > >
> > > ### **Q4: Construction of Multimodal Few-Shot Prompts**
> > >
> > > The prompt is constructed by dynamically assembling four components (Visualized in Fig. 13 & 14):
> > > 1. Task Description: A static text describing the environment and objectives.
> > > 2. Parent Information: The Thought and Code implementation of the selected parent policy.
> > > 3. Behavioral Evidence: Visual feedback generated by the Behavior Summarizer
> > > 4. Instruction: Specific directives for the operator to generate the new policy.
> > >
> > > For implementation details, please refer to get_prompt_m1_M() in supplementary_material/Code/MLES/llm4ad/method/mmeoh/prompt.py, which explicitly demonstrates this assembly process.
> > >
> > > ### **Q5: Acquiring Behavioral Evidence**
> > >
> > > Details are provided in Appendix D.2. Here, we briefly summarize the process:
> > >
> > > - Lunar Lander: We uniformly sample 20 frames from the episode (total ~200 frames) to capture the landing trajectory and attitude adjustments.
> > >
> > > - Car Racing: Instead of raw camera frames, we generate visual trajectory plots based on state history. These plots provide a global view of the racing line and cornering behavior, which allows the MLLM to diagnose issues like "turning too late" or "overshooting" more effectively than raw pixel streams.
> > >
> > > ### **Q6: Evolution of "Thoughts"**
> > >
> > > The "Thought" is a natural language rationale that evolves jointly with the policy. Figure 5 illustrates this evolution.
> > >
> > > There is an example of thought: “The core idea of the new algorithm is to further increase the sensitivity to vertical velocity, making the lander more responsive to abrupt changes, while also lowering the action thresholds to enable quicker decisions for stability control.”
> > >
> > > ---
> > >
> > > We thank the reviewer again for your comments. We will submit the updated paper shortly. We hope our responses and the additional experiments address your concerns, and we would be happy to discuss any remaining questions.

---

> > > > ### Author Response · Authors · 2025-11-28
> > > >
> > > > We sincerely appreciate your constructive suggestions, time, and efforts, as well as your decision to raise the score to 6. Your feedback has significantly helped us improve the quality of our paper.

---

### Official Review · Reviewer_G7L3 · 2025-10-28

**Soundness:** 2
**Presentation:** 2
**Contribution:** 3
**Rating:** 4
**Confidence:** 3

**Summary:**

The black-box nature of neural networks makes typical reinforcement learning policies difficult for humans to interpret. To combat this challenge, previous works employ programs as policies to increase the interpretability. In this work, the authors propose **Multimodal Large Language Model-assisted Evolutionary Search (MLES)**, which leverages visual signals to refine these programs. They introduce a **Behavior Summarizer** to automatically encode the executing trajectories into MLLM-understandable formats for generating new programs. Experimental results indicate that MLES outperform baseline methods, including PPO and **Evolution of Heuristic (EoH)**.

**Strengths:**

- The motivation for including visual signals to improve the generated programs is intuitive and reasonable, as images often consist of richer information than text alone.
- Employing **Python** rather than a **Domain-Specific Language (DSL)** improves the method’s generalization capability, making it more adaptable to various domains and tasks.

**Weaknesses:**

- The experiments are conducted in only two environments, which makes the results less convincing.
- The exclusion of the policy distillation baseline appears insufficiently justified. Although the proposed method is API-based and described as cost-efficient, invoking LLMs around 2000 times is still non-trivial. Distilling a program from a trained policy seems like an approach that could offer a reasonable trade-off between environmental interactions and LLM queries, even if the performance may decline, as the authors state.
- It seems like the main differences between MLES and the previous work EoH lie in the use of the Behavior summarizer and the corresponding Evolutionary operators M1_M and M2_M, which are not clearly stated in the methodology section. It might be better to introduce EoH in a preliminary section and then specify how the proposed method extends it.
- The main contribution of this work lies in the use of the Behavior Summarizer, which is not properly introduced in the main paper but only in the appendix.
- The human evaluation appears somewhat informal; the authors could strengthen it by conducting a more detailed study to examine whether participants’ understanding of the programs truly aligns with their execution outcomes.

**Questions:**

- Do all the evolutionary operators sample programs only once? Given that the temperature is set to 1, sampling multiple programs might introduce sufficient diversity to potentially accelerate the process.
- In Figure 4, the performance curves of PPO and MLES appear to move in a highly synchronized manner. Did the authors observe any underlying reason for this behavior?

---

> ### Author Response · Authors · 2025-11-20
> **Response to Reviewer G7L3 (1/3)**
>
> Thank you for the actionable and constructive feedback. The suggestion to broaden our experimental scope and clarify the methodology significantly strengthens our work. Below, we provide detailed, point-by-point responses.
>
> ### **W1: Broadening Empirical Coverage**
>
> We agree that broader empirical coverage strengthens the paper. To address the concern about "limited environments" and the need for more "convincing results," we conducted two major extensions: (1) validating "from-scratch" policy discovery and (2) evaluating generalization to additional control tasks.
>
> (1) **From-Scratch policy discovery**
> In the original experiments, MLES was provided with an initial executable policy. This may raise the concern that the strong performance stems from the presence of a seed policy. To verify this is not a prerequisite, we conducted new experiments where MLES starts entirely from scratch, operating in two environments with no prior knowledge. The results are shown below:
>
> |                                   | Lunar Lander | Car Racing |
> | :-------------------------------- | :-----------: | :--------: |
> | MLES (with Initial Policy, GPT-4o-mini) |     1.090     |   98.07    |
> | MLES (w/o Initial Policy, GPT-4o-mini)  |     0.667     |   91.96    |
> | MLES (w/o Initial Policy, GPT-5-mini)   |     1.013     |   100.0 (achieved at 209 samples)    |
>
> These results demonstrate that **MLES is fully capable of discovering high-performance policies even without a seed policy**, though it naturally requires more exploration. Notably, when powered by the more capable MLLMs (e.g., GPT-5-mini), MLES achieves excellent performance from scratch while using only one-tenth the number of LLM queries compared to GPT-4o-mini. This illustrates that as MLLMs continue to advance, the effectiveness and applicability of MLES will scale accordingly, enabling it to perform even more robustly across a broader range of tasks and environments.
>
> (2) **Generalization to additional control task**
>
> We fully agree that covering more tasks can further strengthen the paper. We want to reiterate that Lunar Lander and Car Racing were originally chosen to cover a wide range of difficulty (from **discrete action** control to **continuous control**, and from **8-dimensional state inputs** to **high-dimensional image observations**), thereby maximizing empirical diversity within a reasonable experimental scope.
>
> To further evaluate the generalization capability of MLES, we conducted additional experiments on the Inverted Pendulum, specifically designing 10 challenging instances with significant variations in initial angles and positions. The goal was to discover a single programmatic policy capable of balancing the pendulum for 500 steps across all instances.
>
> | Method                      | Inverted Pendulum (avg. steps)|
> | :------------------------- | :---------------: |
> | Initial Policy (PID baselines) |       327.9       |
> | MLES with Initial Policy   |     **500.0** |
> | MLES w/o Initial Policy    |     **500.0** |
>
> MLES successfully discovered a concise, transparent logic that robustly handles all variations, achieving perfect scores. These results will be included in the appendix. We will continue extending MLES to additional tasks and will update the open-source repository accordingly after the double-blind review process, as this is essential for broader community engagement with MLES.

---

> > ### Author Response · Authors · 2025-11-20
> > **Response to Reviewer G7L3 (2/3)**
> >
> > ### **W2: Exclusion of policy distillation baseline**
> >
> > We appreciate this important comment. Our reasoning involves both methodological considerations and practical implications:
> >
> > 1. **Methodological mismatch (inherent vs. post-hoc interpretability)**
> >
> > - Policy distillation typically takes a trained black-box policy (e.g., a high-performing neural net) and fits an interpretable surrogate (program, rule set, decision tree) post hoc. This is a valid and useful line of work (post-hoc interpretability).
> >
> > - MLES, by contrast, aims to directly discover programmatic, human-readable policies during the search process via environment interactions (inherently interpretable policies). The workflow and research question are therefore different: MLES asks whether a multimodal LLM-guided evolutionary loop can produce programmatic policies competitive with DRL, rather than how to compress a trained black-box into code afterwards.
> >
> > 2. **Considerations for Experimental Fairness**
> >
> > - A Distillation comparisons require training high-performance black-box policies first (with their own compute/interaction budgets) and then performing a careful distillation protocol (hyperparameter tuning, multiple seeds) to produce fair baselines. This creates a different experimental pipeline with its own costs and confounders, making a direct apples-to-apples comparison difficult without a substantial additional study.
> >
> > However, we agree that a comprehensive comparison across interpretable modeling paradigms (including distillation) is meaningful and would make an excellent standalone survey. The omission here was intended to maintain methodological coherence. We plan to explore this direction in future work.
> >
> > ### **W3: Improving presentation related to EoH**
> >
> > Thank you for this suggestion. We have reorganized Section 2 to improve clarity:
> >
> > 1. Positioning: We explicitly position MLES as a general automated discovery framework enhanced by multimodal feedback.
> > 2. Relationship between MLES and EoH: We clarify that MLES-EoH serves as an illustrative instantiation rather than a strict dependency. We also make explicit the deliberately minimal modifications we apply relative to EoH, and explain why these choices were made (specifically, to isolate the contribution of multimodal behavioral evidence).
> > 3. Explanation of EoH and distinction from MLES: We provide a clear introduction to the core ideas behind EoH, highlight how the research focus of MLES differs, and show how MLES extends EoH in a principled manner.
> >
> > ### **W4: Improving presentation about Behavior Summarizer**
> >
> > Thank you for highlighting this crucial point. In the original draft, many implementation details were placed in the appendix due to strict page limits.
> >
> > In the revised version, we have reorganized the relevant parts of the Methodology section (Section 2) to ensure the methodology is self-contained:
> >
> > 1. Design Rationale: We added a clearer explanation of why the behavior summarizer is critical for control tasks (Lines 162–165).
> > 2. Workflow: We included a description of the summarizer's workflow within the MLES loop (In lines 275-277).
> > 3. Division of Labor: We explicitly clarified the functional roles of the Behavior Summarizer (and its output, Behavioral evidence).
> >
> > We respectfully invite the reviewer to examine the updated revision and would be happy to further improve the clarity of the presentation based on additional suggestions.

---

> > > ### Author Response · Authors · 2025-11-20
> > > **Response to Reviewer G7L3 (3/3)**
> > >
> > > ### **W5: Strengthening the human evaluation**
> > >
> > > Thanks for your suggestion. We conducted an additional human evaluation focusing on code-behavior alignment. Participants were shown programmatic policies (Python code) alongside several short video clips (trajectory animations) of the agent. They were then asked to identify which specific code segments produced the behaviors observed in the clips and to judge whether these behaviors matched the expectations derived from the corresponding code logic.
> > >
> > > The results were positive: participants successfully mapped specific behaviors (such as 'braking on turns') to their governing code segments, confirming that the generated policies are semantically consistent and understandable. We will include the detailed results of this evaluation in the appendix.
> > >
> > > ### **Question 1**
> > > > Do all the evolutionary operators sample programs only once? Given that the temperature is set to 1, sampling multiple programs might introduce sufficient diversity to potentially accelerate the process.
> > >
> > > Each evolutionary operator is invoked repeatedly during the policy discovery process, and each LLM query samples one policy. Under the population-based evolutionary framework, promising parent policies may be selected multiple times and yield multiple offspring (e.g., receiving multiple applications of the M1_M operator).
> > > Thus, although each operator samples a single offspring per call, the repeated application over generations effectively provides the diversity similar to “sampling multiple programs.”
> > >
> > > ### **Question 2**
> > >
> > > > In Figure 4, the performance curves of PPO and MLES appear to move in a highly synchronized manner. Did the authors observe any underlying reason for this behavior?
> > >
> > > This is an interesting observation. However, we believe the synchronized curves are a coincidence rather than evidence of a deeper connection. In practice:
> > > - PPO exhibits high variance across runs, and its upper and lower envelopes differ significantly from MLES’s performance curves.
> > > - As shown in Figure 3, this synchronized pattern does not appear in other tasks, reinforcing that it is not a systematic or meaningful trend.
> > >
> > > ---
> > >
> > > Again, we sincerely thank the reviewer for the constructive and actionable feedback, especially the suggestions on improving the paper's presentation. We have already uploaded a revised version that reorganizes Section 2, and a final version incorporating all clarifications and the additional experiments will be uploaded shortly. We hope our responses and the additional experiments address your concerns, and we would be happy to discuss any remaining questions.

---

> > > > ### Comment · Reviewer_G7L3 · 2025-11-26
> > > > **Thank you for the replies (score raised)**
> > > >
> > > > I thank the authors for the detailed replies. Since my concerns have been resolved, I have raised my score accordingly.

---

> > > > > ### Author Response · Authors · 2025-11-28
> > > > >
> > > > > We sincerely appreciate your constructive suggestions, time, and efforts, as well as your decision to raise the score to 6. Your feedback has significantly helped us improve the quality of our paper.

---

### Official Review · Reviewer_BFU7 · 2025-11-01

**Soundness:** 3
**Presentation:** 3
**Contribution:** 2
**Rating:** 2
**Confidence:** 3

**Summary:**

This paper proposes MLES, a framework that combines multimodal large-language models (MLLMs) and evolutionary computation to automatically discover *programmatic*, interpretable control policies. Unlike prior work using LLMs for reward shaping or heuristic tuning, MLES directly synthesizes executable code policies, enhanced by visual behavioural evidence that guides refinement.

The framework iteratively evolves policies represented as code + “thought” (rationale) + quantitative metrics + behavioural evidence, aiming to achieve human-readable, verifiable control logic.

Experiments on two benchmarks (Lunar Lander and Car Racing) show that MLES reaches performance comparable to Proximal Policy Optimization (PPO) while producing transparent and explainable policies. Ablation studies highlight the contribution of visual feedback and multimodal reasoning to the policy-search efficiency.

**Strengths:**

- The integration of visual behaviour-feedback into the policy-synthesis pipeline is a genuinely novel contribution, enabling richer signals beyond scalar metrics in MLLM-policy generation.
- The paper is well-written and clearly laid out, with the framework, methodology, and experiments described in a comprehensible manner.
- The experiments include a transparent evolution process, showing how policies evolve across generations and how visual evidence influences the search. I personally liked the presentation of Figure 5.
- The authors provide the final generated programmatic policies (code) for the benchmark tasks, enabling inspection and verification of their results.
- The approach successfully balances interpretability (through programmatic policies) with competitive performance (comparable to a strong baseline), which is notable.

**Weaknesses:**

- The novelty is limited: the proposed method seems to be a direct modification of existing work EoH by including visual clues. In my opinion, simply adding visual cues does not count as a major contribution—rather, it relies heavily on the intrinsic ability of the MLLM.
- The experiments are limited and the potential for generalization is questionable. The authors test only on two tasks (Lunar Lander and Car Racing), which are relatively simple and for which simple code-based policies might exist. In more sophisticated scenarios (e.g., inverted-pendulum), it is unclear that the proposed method will still work. Is the pipeline able to start from scratch, or must it be provided with a PID-like physical model so that the LLM finds parameters?

**Questions:**

1. Could the authors apply their pipeline to tasks like the inverted pendulum and ask LLM for a hard-coded policy, or for finding the optimal parameters of a PID controller? Would the proposed pipeline still succeed in both cases?
2. How dependent is the method on the quality of the MLLM used? What happens if a smaller or less capable multimodal model is used? How often does the pipeline require manual intervention?
3. Are there limitations when the environmental dynamics are high-dimensional, noisy, or partially observable? How does the method scale in those settings?

---

> ### Author Response · Authors · 2025-11-20
> **Response to Reviewer BFU7 (1/3)**
>
> We sincerely thank the reviewer for your valuable time and constructive feedback. We are encouraged that the reviewer recognized the "genuinely novel contribution" of integrating visual feedback and found the paper clearly written. Below, we respond to Weakness 1 (W1), Weakness 2 (W2), and the three Questions in order.
>
> ### **W1: Clarification on Novelty, Methodological Choices, and Broader Impact**
>
> We sincerely appreciate the reviewer's feedback, which indicates that our original presentation may have undersold the innovation. To clarify our novelty and contributions, we organize our response into four focused points:
>
> **(1) Far from "Simple Add-on": A Meaningful Shift in Search Paradigm**
>
> We want to clarify that MLES is not a "simple modification." Rather, MLES represents a meaningful shift in the LES paradigm—**from semi-blind, scalar-driven search to targeted, rationale-driven refinement**.
>
> - Prior LES work (like EoH) relies purely on scalar rewards. A value (e.g., -150) tells the LLM that the policy failed, but **provides no information as to *Why***. This ambiguity can lead to random, unguided policy mutations and inefficient exploration, as shown in Fig. 3 and Fig. 4.
>
> - In contrast, MLES leverages a high-bandwidth visual feedback, enabling the MLLM to perform qualitative, human-like behavioral critique and causal credit assignment, leading to targeted and intelligent policy modifications. For example, as shown in Fig. 5, the operator with visual feedback can provide BE-driven analysis like "Frequent loss of control in sharp turns due to high speed." This marks an important shift toward **interpretable, grounded policy evolution**.
>
> **(2) Deliberate minimal modification to EoH to scientifically isolate the contribution of visual feedback**
>
> Our choice to build upon EoH with minimal changes was intentional. Our primary claim centers on the benefits of ***visual feedback***. By introducing visual feedback as a **minimal modification to a recognized baseline**, we can scientifically isolate its contribution and more clearly demonstrate the performance gains.
>
> If we had instead introduced multiple new components (e.g., complex EC operators, different search strategies), it would have conflated the sources of improvement. By keeping the EC framework stable, we rigorously demonstrate that the visual feedback mechanism alone drives the performance gains.
>
> **(3) Operationalizing Multimodal Reasoning in Automated Discovery**
>
> We agree that our MLES leverages the "intrinsic ability" of the MLLM, but we argue that this is a strength and a meaningful contribution. Although MLLMs have existed for some time, the community still lacks effective approaches that channel their multimodal reasoning for automated policy/algorithm discovery. Our contribution lies in **designing an approach that successfully operationalizes MLLM's multimodal reasoning within an automated search loop**. We believe this provides a concrete and generalizable insight into how MLLMs can be effectively integrated into computational search. Existing LES methods can adopt our technique to enhance their frameworks, similar to how it integrates with EoH, which is beneficial for the LES field.
>
> **(4) Broader Impact**
>
>    Our work offers contributions across multiple communities:
>    - **Control Policy Discovery:** We introduce a new approach that achieves performance comparable to strong DRL (e.g., PPO) while producing *fully transparent, human-readable programmatic policies*.
>    - **LES (LLM-assisted Evolutionary Search):** To the best of our knowledge, this is among the ***first*** successful attempts to integrate multimodal feedback into the LES loop, demonstrating clear advantages over text-only, scalar-based variants.
>    - **Evolutionary Computation (EC):** We empirically validate that MLLMs can function as *high-level, reasoning-driven operators* in evolutionary search, providing human-like behavioral insights to guide the evolution process.
>
> In addition, many recent works in the LES field aim to improve performance through increasingly sophisticated prompt engineering or by incorporating mature EC techniques— directions that are complementary to ours. **Our work explores a fundamentally different dimension**. Instead of optimizing the prompting or the evolutionary operators themselves, we focus on enhancing LES by improving the quality and bandwidth of information available during the evolutionary loop. By leveraging multimodal reasoning, **MLES enables each evolutionary step to become more targeted, interpretable, and justifiable**. We sincerely hope that this perspective offers a fresh and valuable contribution to the LES community.

---

> > ### Author Response · Authors · 2025-11-20
> > **Response to Reviewer BFU7 (2/3)**
> >
> > In the revised manuscript, we reorganized Section 2 to present our work more clearly:
> > 1. We explicitly position MLES as a **general automated discovery framework enhanced by multimodal feedback**, highlighting its conceptual advance.
> > 2. We clarify that MELS-EoH serves as an illustrative instantiation rather than a strict dependency. We also make explicit the deliberately minimal modifications we apply relative to EoH, and explain why these choices were made (specifically, to isolate the contribution of multimodal behavioral evidence).
> > 3. We provide a clear introduction to the core ideas and design intentions behind MLES and EoH, and show how MLES extends EoH in a principled manner.
> >
> > We respectfully invite the reviewer to examine the updated revision and would be happy to further improve the clarity of the presentation based on additional suggestions.
> >
> > ---
> >
> > ### **W2(1) & Q1: Clarification on Benchmarks and Generalization to More Sophisticated  Scenarios (e.g., Inverted Pendulum)**
> >
> > The Lunar Lander and Car Racing environments cover a range from **discrete-action** to **continuous-control**, and **8-dimension** to **high-dimensional image observation spaces**. While the tasks are **standard** benchmarks, they are **not simple**. Existing code-based policies (like the Initial Policy in Table 1) fail to achieve good performance. Moreover, the policies discovered by MLES are not "simple." For instance, the final Car Racing policy (Appendix, Fig. 21) is a non-trivial program with adaptive throttle management, speed-based steering clamps, and anti-stalling logic.
> >
> > To further evaluate the generalization capability of MLES, we conducted additional experiments on **Inverted Pendulum**, specifically designing 10 challenging instances with significant variations in initial angles and positions. The goal was to discover a single programmatic policy capable of balancing the pendulum for 500 steps across all instances.
> >
> > | Method                      | Inverted Pendulum (avg. steps)|
> > | :------------------------- | :---------------: |
> > | Initial Policy (PID baselines) |       327.9       |
> > | MLES with Initial Policy   |     **500.0** |
> > | MLES w/o Initial Policy    |     **500.0** |
> >
> > As shown, **MLES extends effectively to the Inverted Pendulum setting** and achieves the maximum reward both with and without an initial policy. We believe this addresses the reviewer’s concerns regarding the generalization ability of our MLES.
> >
> > ### **W2(2): MLES's Ability to Start from Scratch**
> >
> > To empirically address the concern about dependence on an initial policy (seed), we conducted additional experiments starting from scratch (without any initial policy) over 2000 LLM calls:
> >
> > |                                   | Lunar Lander | Car Racing |
> > | :-------------------------------- | :-----------: | :--------: |
> > | MLES (with Initial Policy, GPT-4o-mini) |     1.090     |   98.07    |
> > | MLES (w/o Initial Policy, GPT-4o-mini)  |     0.667     |   91.96    |
> > | MLES (w/o Initial Policy, GPT-5-mini)   |     1.013     |   100.0    |
> >
> > Results show that while starting from scratch requires more exploration, **MLES is fully capable of discovering high-performance policies**. Notably, when powered by the more capable GPT-5-mini, MLES achieves excellent results even without an initial policy. The ability to ingest an initial policy is an **optional advantage** to accelerate search, **not a strict requirement**. We hope this clarification addresses the reviewer's concerns about generalization.

---

> > > ### Author Response · Authors · 2025-11-20
> > > **Response to Reviewer BFU7 (3/3)**
> > >
> > > ### **Q1: Applicability to PID or hard-coded policies**
> > > > Could the authors apply their pipeline to tasks like the inverted pendulum and ask LLM for a hard-coded policy, or for finding the optimal parameters of a PID controller? Would the proposed pipeline still succeed in both cases?
> > >
> > > As shown in our response to **W2(1) & Q1**, we have successfully applied MLES to the **Inverted Pendulum** task (both from scratch and from an existing PID policy), which should address the first part of the reviewer's concern.
> > >
> > > Furthermore, we wish to emphasize that MLES focuses on evolving the ***control logic*** (code structure) itself, rather than only tuning parameters. While it could be adapted to tune parameters for a fixed structure (e.g., PID controller), this would underutilize its core strength. MLES’s main contribution is discovering novel, human-readable control logic, providing a new paradigm for transparent and verifiable control policy development, as demonstrated by the Inverted Pendulum experiments both from scratch and from an initial policy.
> > >
> > > ### **Q2(1): Dependence on MLLM quality**
> > >
> > > As we discuss in Appendix A.3, “The performance ceiling of MLES is inherently coupled with the capabilities of its underlying MLLMs. The framework’s ability to understand complex failure modes is contingent on the MLLM’s visual reasoning, while the quality of generated policies depends on its code generation and logical deduction abilities.” Like all LES paradigms[1, 2], MLES's capability is influenced by the underlying MLLM.
> > >
> > > Below is a comparison using GPT-5o-mini (released Aug 2025) and GPT-4o-mini (released June 2024).
> > >
> > > |  | Lunar Lander     |    |       Car Racing      |     |
> > > |--|:----:|:----:|:---:|:----:|
> > > | | Score at 2000 samples | Samples to reach 4o-mini's 2000-sample score | Score at 2000 samples | Samples to reach 4o-mini's 2000-sample score |
> > > | MELS with Initial Policy (GPT-4o-mini) | 1.09  | -  | 98.07 | - |
> > > | MELS with Initial Policy (GPT-5-mini)  |  1.065  | 2000 | 100  | 346 |
> > > | MELS w/o Initial Policy (GPT-4o-mini)  | 0.667 | - | 91.96   | -  |
> > > | MELS w/o Initial Policy (GPT-5-mini)   | 1.013 | 258| 100| 209|
> > >
> > > More powerful MLLMs bring stronger (and faster) performance to MLES. Conversely, a smaller or less capable model would reduce performance. However, we stress that even the previous generation of MLLMs (GPT-4o-mini) is already sufficient for MLES to perform excellent policy discovery. As MLLMs continue to evolve, MLES, as a framework, will naturally benefit from lower costs and stronger performance.
> > >
> > > [1] Liu, Fei, et al. "Evolution of heuristics: Towards efficient automatic algorithm design using large language model." arXiv preprint arXiv:2401.02051 (2024).
> > > [2] Hu, Qinglong, and Qingfu Zhang. "Partition to Evolve: Niching-enhanced Evolution with LLMs for Automated Algorithm Discovery." The Thirty-ninth Annual Conference on Neural Information Processing Systems (Neurips 2025).
> > >
> > > ### **Q2(2): No manual intervention is required**
> > >
> > > Our policy discovery pipeline is fully automated; no manual intervention is required throughout the process.
> > >
> > > ### **Q3: Scaling to high-dimensional, noisy, or partially observable environments**
> > >
> > > We acknowledge that our current experiments did not address noisy or partially observable settings. This is a key challenge for future work, particularly for real-world tasks like autonomous driving.
> > >
> > > Potential solutions are discussed in **Appendix A.1**. The policies generated by MLES exhibit clear **modularity**, which opens up the possibility of integrating **advanced perception modules** (like pre-trained CNNs) to handle high-dimensional or noisy environments. For partially observable settings, extending MLES to generate policies with **memory** (e.g., simple filters or belief updates) is a promising direction for future work.
> > >
> > > These are insightful points, and we will highlight them as key areas for future exploration in the revised manuscript.
> > >
> > > ---
> > > ### **Summary**
> > >
> > > MLES offers a new perspective, demonstrating strong performance and generalization. It can design policies **from scratch and excels at leveraging existing prior knowledge**. Its performance will naturally scale with the improvement of MLLMs, and there are clear avenues for extension to more complex environments.
> > >
> > > We thank the reviewer again for the comments. We are confident that our work possesses sufficient novelty and provides valuable insights to the community. We will revise the manuscript accordingly and welcome any further discussion to elaborate on any remaining points. We hope our clarifications assist in reassessing the evaluation.

---

> ### Author Response · Authors · 2025-11-27
> **Gentle reminder regarding our response and new experiments (Submission 17263)**
>
> Dear Reviewer BFU7,
>
> We sincerely appreciate the time and effort you have dedicated to reviewing our paper. We wanted to gently check if you have had a chance to review our response.
>
> Following your suggestions, we have completed the additional experiments on the Inverted Pendulum and the "start-from-scratch" setting, providing strong evidence to demonstrate the generalizability of our method. We have also revised our manuscript to reflect these updates and to further clarify the novelty and impact of this work.
>
> Should you have any further points or questions, we are more than willing to engage in additional discussion to further improve the quality of our paper.
>
> Best regards,
>
> The Authors of Submission 17263

---

> > ### Comment · Reviewer_BFU7 · 2025-11-28
> >
> > I thank the authors for the detailed rebuttal, the additional experiments on the Inverted Pendulum, and the clarifications regarding the design of MLES. The new results (including the “from scratch” setting) do help show that the pipeline can complete more complex control tasks.
> >
> >
> >
> > Here are some remaining points that are unclear to me.
> >
> >
> >
> > **1. Scope of tasks and strength of claims**
> >
> >
> >
> > All evaluated environments—Lunar Lander, Car Racing, and now Inverted Pendulum—are still relatively simple, fully observed classical control benchmarks for which short, hand-engineered Python policies are known or plausible. This makes them well aligned with the current strengths of LLMs (code + basic visual reasoning), but also means the empirical evidence remains restricted to a narrow family of toy problems.
> >
> >
> >
> > In particular, the Inverted Pendulum setting is still a low-dimensional, rigid-body system. It does not yet demonstrate that the framework can handle more realistic, high-dimensional, noisy, or partially observable dynamics, where purely programmatic policies may be harder to express and discover. For this reason, I remain cautious about some of the broader “paradigm shift”–style claims; the current method looks closer to “EoH + multimodal rollouts” than a fundamentally new algorithmic framework. I would encourage the authors to substantially tone down such claims and clearly delimit the demonstrated scope.
> >
> >
> >
> > **2. Request on Inverted Pendulum details and templates**
> >
> >
> >
> > Since the rebuttal relies heavily on the new Inverted Pendulum results as evidence of generalization, it becomes important that this part is fully inspectable. It would be helpful if the authors provided the following:
> >
> >
> >
> > - the **full generated programmatic policies for Inverted Pendulum** in the appendix or supplementary material,
> >
> > - the **exact prompts** used for these experiments, and
> >
> > - the **code templates** used in both settings:
> >
> >   (i) with an initial PID-like policy, and
> >
> >   (ii) without any initial policy (“from scratch”).
> >
> >
> >
> >
> >
> > This would help readers judge how much structure comes from templates vs. being synthesized, and whether the resulting policies are genuinely novel or mostly re-express known control logic.
> >
> >
> >
> > **3. Question on potential training-data contamination**
> >
> >
> >
> > I also have a conceptual concern that I encourage the authors to address explicitly. All considered tasks are standard benchmark environments (including Inverted Pendulum), for which many canonical implementations and heuristic controllers exist online. Given that the underlying MLLMs are trained on large-scale internet data, it seems quite plausible that standard control code for these problems appears, directly or indirectly, in the training corpus.
> >
> >
> >
> > This raises an important question:
> >
> >
> >
> > > To what extent can we be confident that MLES is *discovering* new programmatic policies, rather than reproducing and lightly adapting control code that the model may have effectively memorized?
> >
> >
> >
> > It would be helpful if the authors could help rule out this possibility (e.g., by checking for near-duplication against canonical implementations, or by using task variations that reduce the likelihood of direct memorization).

---

> ### Author Response · Authors · 2025-11-29
> **Response to Reviewer BFU7 (Follow-up, Part 1 of 2)**
>
> We thank the reviewer for the prompt follow-up and for acknowledging that our new experiments demonstrate the pipeline's generalization and capability on complex tasks.
>
> Below, we provide the requested details and clarifications to fully resolve the remaining concerns.
>
> ### **1. Response to Point 1 (1/2): Clarifying the "Paradigm Shift" and Claims**
>
> We clarify that the term "Search Paradigm Shift" refers to a fundamental transformation **in the nature of the policy generation process**, rather than an abandonment of the underlying "EC + LLM" framework.
>
> We respectfully maintain that our claims are accurate and well-supported by our results. As stated in our revision (Lines 79-84):
>
> > “... MLES introduces visual feedback-driven behavior analysis into the policy generation pipeline. By leveraging MLLMs to scrutinize execution traces beyond mere scalar metrics, MLES enables targeted, rationale-driven policy refinement. This approach fundamentally transforms LLM-driven automated discovery **from stochastic trial-and-error into a grounded, diagnostic refinement process**, rendering the evolution transparent, traceable, and highly efficient.”
>
> We provide strong evidence to validate this specific claim:
>
> - **Unique Transparency and Traceability** (Fig. 5): Unlike prior LES methods that **can only** present iterative score curves and modified policy, MLES provides the **reasoning behind each policy modification**. Figure 5 demonstrates that MLES explicitly diagnoses ***Why*** a policy failed before fixing it. This level of transparency and traceability is a capability that purely scalar-based LES methods fundamentally lack.
>
> - **Efficiency Gains** (Figs. 3 & 4): The shift from "guessing" to "diagnosing" directly translates to performance. Our results show MLES achieves consistently and significantly higher search efficiency compared to EoH.
>
> We believe that with this clarification regarding "Shift", the concern regarding "overclaiming" is resolved.
>
> **Clarification on MLES Independence:**: MLES does not depend on EoH; rather, it is a general framework. By modifying the evolutionary operators and parent selection mechanisms, it can be made **compatible with any LES method**. Our decision to instantiate MLES on top of EoH was to provide a scientifically controlled demonstration of our claims.
>
> ---
> ### **2. Response to Point 1 (2/2): Justifying the Task Scope and Experimental Setting**
>
> We address the concerns regarding the scope of tasks with the following points, demonstrating that our setting is appropriate, sufficient, and rigorous:
>
> **(1) Correction on Dimensionality**: We respectfully clarify that Car Racing **is not a low-dimensional task** regarding its observation space. It requires processing high-dimensional image inputs ($96 \times 96 \times 3$) to extract state information. The success of MLES here proves its capability to handle high-dimensional visual perception.
>
> **(2) Existence ≠ Simple**：The reviewer suggests the tasks are simple because hand-engineered policies "exist." This reasoning is akin to claiming that "NP-hard problems are easy simply because a solution can be easily defined and verified". **We respectfully argue that the existence of a known solution does not imply that the problem is easy to solve**.
>
> Actually, our extensive survey of open-source solutions reveals that while simple heuristics exist (used as "Initial Policies" in Table 1), they are merely "executable", often fragile and sub-optimal. MLES automatically discovers robust policy that rivals PPO performance under the same rollouts, demonstrating its excellent performance.
>
> **(3) Sufficiency and Completeness of Validation**: We argue that the chosen tasks, combined with the new experiments during the rebuttal period, are complete and sufficient to validate our core contributions:
>
> 1. **Methodological Advance**: All results show that MLES comprehensively outperforms EoH, demonstrating that *visual feedback-driven behavior analysis* renders the evolution transparent, traceable, and highly efficient.
> 2. **Performance**: MLES **matches strong DRL baselines (PPO)** while maintaining the benefits of programmatic transparency.
> 3. **Generalization**: MLES successfully discovers excellent policies **from scratch** across multiple distinct tasks.
> 4. **Knowledge Transfer**: The "from-scratch" setting demonstrates that MLES overcomes the limitations of predefined domain-specific languages, while the "initial policy" setting demonstrates its capacity for efficient knowledge reuse.
>
> **Regarding Noisy and Partially Observable Environments**: While MLES is currently demonstrated on fully observed tasks, this does not undermine our core claims. The success on current benchmarks demonstrates the potential for handling more complex dynamics. As discussed in Appendix A.1, the modular nature of MLES policies allows for future integration with established perception **tools** or **modules** to handle noisy or partially observable environments.

---

> > ### Author Response · Authors · 2025-11-29
> > **Response to Reviewer BFU7 (Follow-up, Part 2 of 2)**
> >
> > ### **3. Response to Point 2: Full Transparency on Inverted Pendulum (Details and Templates)**
> >
> > We thank the reviewer for the suggestion. To ensure full inspectability and reproducibility, we have included all requested materials in the **Appendix G** of the revised manuscript. Specifically, we have added:
> > 1. Full Generated Policies: The complete code for the best-evolved Inverted Pendulum policies.
> > 2. Exact Prompts: The prompts used for the experiments.
> > 3. Code Templates: The template used for both the "from scratch" and "from initial policy" settings.
> >
> > We have also updated **the supplementary material with the corresponding experimental code**. We believe these additions fully address the reviewer's concerns regarding transparency.
> >
> >
> > ### **4. Response to Point 3: Addressing Concern Regarding Potential Data Contamination**
> >
> > The reviewer raises a question:
> > > Whether MLES is simply retrieving memorized code from the MLLM's training data.
> >
> > We are **fully confident** that MLES performs active search and reasoning, rather than merely retrieving memory.
> >
> > We provide two distinct lines of evidence to support this：
> >
> > **(1) Evidence from Evolutionary Trajectory**
> > If the MLLM were "retrieving" control codes from its training data, the performance curves in Figures 3 and 4 would exhibit a "step function", i.e., jumping to the optimal score immediately in Generation 0 or 1.
> >
> > **Reality**: Our results show a clear, gradual evolutionary curve. MLES starts with sub-optimal performance and **improves iteratively**. This progressive trend reflects a genuine process of diagnosis and repair, where the model analyzes specific failure modes to refine the policy, rather than reciting static answers.
> >
> > **(2) Evidence from the Performance Gap (Visual-enhanced vs. Text-only)**
> >
> > This is the strongest evidence. We used the same underlying model (GPT-4o-mini) for both MLES and the baseline EoH. If the model were relying on memorization (e.g., "adapting control code from the internet"), it would not need visual feedback to do so. The text-based EoH should perform equally well.
> >
> > **Reality**: MLES significantly outperforms EoH in both efficiency and final score. The performance gap proves that the model cannot solve the task via simple retrieval. Instead, the visual feedback enables the MLLM to reason about the specific dynamics of the current policy, leading to targeted policy refinement.
> >
> > These two points prove that MLES operates as an active reasoning agent that synthesizes policies based on real-time observational evidence, rather than simple memory retrieval.
> >
> > ---
> >
> > Thank you again for your time and effort in reviewing our paper. Throughout our exchanges, your insightful questions and suggestions have notably enhanced the clarity and empirical rigor of our work. The revised version has been uploaded to reflect these improvements.
> >
> > We believe that our detailed responses to all raised points, combined with the new experimental evidence, thoroughly address your remaining concerns and provide a solid basis for reassessing your evaluation of our paper.

---

### Author Response · Authors · 2025-11-30
**Authors' Summary to Area Chair**

Dear Area Chair,

Thank you for your time and effort in managing our submission. In light of the unexpected incident and following the chairs’ instructions, we respectfully provide a concise summary to assist your final assessment. This summary includes (1) the scientific value of the work, (2) our substantial efforts during the rebuttal, and (3) a factual overview of the discussion process.

---

### **1. Scientific Contribution and Significance**

This work introduces MLES, a general framework that leverages the multimodal reasoning capabilities of MLLMs to systematically enhance the LES paradigm for efficiently discovering programmatic policies. We highlight two contributions that we believe are meaningful for the community:

**(1) Bridging high performance and interpretability in policy discovery**

MLES is empirically verified to generate policies that achieve competitive performance with the strong DRL methods. **Crucially**, unlike the opaque neural networks typical of DRL, the resulting policies from MLES are human-readable programs, offering fine-grained transparency and verifiability of control logic. This provides a meaningful alternative paradigm for designing trustworthy control policies.

**(2) Introducing behavior-oriented evolution to the LES paradigm**

Many recent LES efforts primarily *refine prompt engineering or combine EC techniques* to improve efficiency. In contrast, MLES introduces a *visual-feedback–driven behavior analysis* mechanism in the policy generation process, enabling MLLMs to **identify the causes of failure** and **perform targeted corrections** to yield better policies. This shifts automated discovery **from stochastic trial-and-error into a grounded, diagnostic refinement process**, rendering the evolution transparent, traceable, and highly efficient. We believe this methodological shift is novel and beneficial for the LES community.

---
### **2. Substantive Rebuttal Efforts**

During the rebuttal period, we systematically addressed all reviewer comments by conducting additional experiments and expanding detailed explanations. We are confident we have not overlooked any of the reviewers' concerns. Our main efforts include:

**(1) Manuscript improvements for clarity and quality**
- **Revised Figure 1** to ensure it is fully self-contained.
- **Clarified novelty** by explicitly highlighting the unique contributions of MLES over standard LES.
- **Clarified MLES positioning**, emphasizing that MLES is a general framework, while MLES-EoH is an instantiation used for scientific validation.
- **Expanded methodological details**, adding descriptions of the Behavior Summarizer, selection pressure, and multimodal few-shot prompt construction.

**(2) Expanded Empirical Validation**
- **Generalization**: Added “From Scratch” experiments across all tasks and extended MLES to the Inverted Pendulum task.
- **MLLM dependency Analysis**: Conducted ablation comparing GPT-4o-mini and GPT-5-mini to study effifiency/performance gaps.

---
### **3. Factual Overview of the Discussion Process**

Before Nov. 27, we had already reached a clear consensus with three reviewers (**EJft, qyrE, G7L3**). They **explicitly confirmed that their concerns were resolved by the rebuttal** and **raised ratings to 8, 6, and 6, respectively**. Reviewer **BFU7** acknowledged our newly added experiments, and the discussion was ongoing when the freeze occurred.  We believe our latest response sufficiently addresses the remaining concerns of Rev. **BFU7**, and we respectfully invite your assessment. Below is a factual reconstruction of the interactions:

|Reviewer|Author-Reviewer Interaction Before Leak Notification (Before Nov 27)|Status at Notification Moment|Interaction After Notification|
|:-|:-|:-|:-|
|**EJft**|① Author response (19 Nov, 22:45); ② Reviewer 2nd-round comments (25 Nov, 09:43); ③ Author 2nd-round response (26 Nov, 19:25);  **④ Reviewer CONFIRMED resolution & Raised Score 6 → 8** (26 Nov, 21:14)|**✅ Consensus Reached** (Rating: **8**)|—|
|**qyrE**| ① Author response (17 Nov, 16:50); **② Reviewer Raised Score 4 → 6 (No further comments)** (23 Nov, 05:33)|**✅ Consensus Reached** (Rating: **6**)|—|
|**G7L3**| ① Author response (20 Nov, 14:17); **② Reviewer CONFIRMED resolution & Raised Score 4 → 6** (26 Nov, 16:12)|**✅ Consensus Reached** (Rating: **6**)|—|
|**BFU7**|① Author response (20 Nov, 15:10)|**⏳ Discussion Active** (Pending Resolution)|**② Reviewer 2nd-round comments** (28 Nov, 17:16); **③ Author 2nd-round response** (29 Nov, 23:37)|

**Evidence**: We have uploaded a *snapshot of the scores* and *the full timestamped interaction logs* (Rebuttal Page before system revert) in the **Supplementary Material** for your verification. We ensure that all provided materials strictly adhere to double-blind policies.

---

We sincerely thank the Area Chairs and Reviewers for their dedication. The constructive feedback has helped us substantially refine the clarity and presentation of our submission.

---

### Meta-Review · Area_Chair_KbTL · 2026-01-01

**Summary:**

This paper proposes MLES, a framework for discovering programmatic, human-readable control policies by integrating multimodal large language models into an evolutionary search loop. The key idea is to augment traditional scalar-reward–driven LLM-assisted evolutionary search with visual, behavior-level feedback, allowing the model to diagnose failure modes from rollouts and apply targeted, rationale-driven policy modifications. The framework is instantiated on top of an existing LES method and evaluated on Lunar Lander, Car Racing, and—added during rebuttal—Inverted Pendulum, including “from-scratch” settings without seed policies. Results show that MLES achieves performance comparable to PPO while producing transparent code policies and more sample-efficient search than text-only baselines.

The main strengths of the work are the clear motivation toward interpretability, the technically sound integration of multimodal behavioral evidence into the evolutionary loop, and the strong rebuttal that added substantial new experiments, clarified novelty relative to prior LES work, and improved methodological transparency. In particular, the added Inverted Pendulum experiments, ablations on starting from scratch, and analyses of MLLM dependence directly addressed concerns about generality and over-reliance on initial policies. Weaknesses remain in the limited scope of evaluated environments and the fact that all tasks are classical, fully observed benchmarks that align well with current LLM capabilities; broader claims about paradigm shifts therefore rely partly on conceptual argument rather than empirical coverage. Nevertheless, post-rebuttal discussion indicates that most reviewers viewed these limitations as acceptable for the paper’s stated contributions, especially after claims were clarified and additional materials were provided.

Overall, I believe this paper should be accepted.

**Reviewer Concerns:**

Several major concerns raised in the initial reviews were convincingly addressed by the rebuttal. These include doubts about novelty relative to prior LLM-assisted evolutionary search, dependence on a seed policy, and lack of generalization evidence beyond the original two environments. The authors clarified that MLES is a general framework rather than a minor tweak, justified the minimal modification strategy to isolate the effect of multimodal feedback, and demonstrated meaningful efficiency gains over text-only baselines. Crucially, new experiments demonstrated that MLES can discover effective policies from scratch and handle an additional control task (Inverted Pendulum), directly addressing questions about generalization and initialization. Requests for transparency were also addressed by adding fully generated policies, prompts, and templates to the appendix and supplementary materials.

Some concerns remain partially outstanding. The empirical scope remains restricted to relatively simple, fully observed control benchmarks, leaving open questions about scalability in noisy, partially observable, or more realistic domains. Relatedly, skepticism persists about whether the demonstrated gains justify broader “paradigm shift” language, even though the authors substantially clarified and narrowed their claims. Finally, while the authors provided reasonable arguments and evidence against simple memorization from training data, definitively excluding this possibility in standard benchmarks is inherently difficult and remains a conceptual limitation rather than a resolved issue.

**Reviewer Scores:**

Taken as a whole, the reviews indicate an upward shift in scores following a full consideration of the rebuttal. Reviews that were initially positive or borderline became clearly supportive once additional experiments and clarifications were provided, moving from marginal or tentative acceptance into solid acceptance territory. Even the initially negative assessment acknowledged after the rebuttal that the new results demonstrated increased task coverage and pipeline capability, and the remaining reservations were framed more as scope limitations than fundamental flaws. Overall, the post-rebuttal consensus trends toward acceptance, with the distribution of opinions aligning more closely around a positive evaluation than the initial score spread suggested.

---

### Decision · Program_Chairs · 2026-01-26

Accept (Poster)